# Combining a pollen and macrofossil synthesis with climate simulations for spatial reconstructions of European climate using Bayesian filtering

Nils Weitzel[1,2], Andreas Hense[1], and Christian Ohlwein[1]

[1]Institut für Geowissenschaften und Meteorologie, Rheinische Friedrich-Wilhelms-Universität Bonn, Auf dem Hügel 20, 53121 Bonn, Germany
[2]Institut für Umweltphysik, Ruprecht-Karls-Universität Heidelberg, Im Neuenheimer Feld 229, 69120 Heidelberg, Germany

**Correspondence:** Nils Weitzel (nils.weitzel@uni-bonn.de)

**Abstract.** Probabilistic spatial reconstructions of past climate states are valuable to quantitatively study the climate system under different forcing conditions because they combine the information contained in a proxy synthesis in a comprehensible product. Unfortunately, they are subject to a complex uncertainty structure due to complicated proxy-climate relations and sparse data, which makes interpolation between samples difficult. Bayesian hierarchical models feature promising properties to
handle these issues like the possibility to include multiple sources of information and to quantify uncertainties in a statistically rigorous way.

We present a Bayesian framework that combines a network of pollen and macrofossil samples with a spatial prior distribution estimated from a multi-model ensemble of climate simulations. The use of climate simulation output aims at a physically reasonable spatial interpolation of proxy data on a regional scale. To transfer the pollen data into (local) climate information,
we invert a forward version of the probabilistic indicator taxa model. The Bayesian inference is performed using Markov chain Monte Carlo methods following a Metropolis-within-Gibbs strategy.

Different ways to incorporate the climate simulations in the Bayesian framework are compared using identical twin and cross-validation experiments. Then, we reconstruct mean temperature of the warmest and mean temperature of the coldest month during the mid-Holocene in Europe using a published pollen and macrofossil synthesis in combination with the Paleo-
climate Modelling Intercomparison Project Phase III mid-Holocene ensemble. The output of our Bayesian model is a spatially distributed probability distribution that facilitates quantitative analyses which account for uncertainties.

## 1 Introduction

Spatial or climate field reconstructions of past near surface climate states combine information from proxy samples, which are mostly localized, with a model for interpolation between those samples. They are valuable for comparisons of the state of
the climate system under different external forcing conditions, because they produce a comprehensible product containing the joint information in a proxy synthesis. Thereby, spatial reconstructions are more suitable for many quantitative analyses of past climate than individual proxy records. Unfortunately, spatial reconstructions are subject to a complex uncertainty structure due

to uncertainties in the proxy-climate relation and the sparseness of available proxy data which leads to additional interpolation uncertainties. Therefore, a meaningful reconstruction has to include these uncertainties (Tingley et al., 2012). A natural way to represent uncertainties in the proxy-climate relation are so-called probabilistic transfer functions (Ohlwein and Wahl, 2012). To account for the uncertainties due to sparseness of proxy data, we suggest the use of stochastic interpolation techniques. Most standard geostatistical methods like kriging or Gaussian modelling with Matérn covariances are designed for interpolation in data rich situations, while in paleoclimatology we deal with sparse data. Therefore, their direct application to paleo situations is not suitable. Instead, we propose to use interpolation schemes that contain additional physical knowledge, such that the resulting product combines the information from a proxy network in a physically reasonable way (Gebhardt et al., 2008). Our approach can in addition be used for structural extrapolation of the proxy data.

We use Bayesian statistics to combine the two modules mentioned above: The (local) proxy-climate relation and spatial interpolation. The Bayesian framework allows the combination of multiple data types. In our case, these are pollen and macro-fossil records to constrain the local climate, and climate simulations, which produce physically consistent spatial fields for a given set of large scale external forcings. In addition, our framework accounts for several sources of uncertainty in a statistically rigorous way by estimating and inferring a multivariate probability distribution, the so-called posterior distribution (Gelman et al., 2013).

Pollen are the terrestrial proxy with the highest spatial coverage (Bradley, 2015), and there is a long tradition of using them for inferring past climate by applying statistical transfer functions (Birks et al., 2010). In recent years, several traditional transfer functions like indicator taxa, modern analogues, and weighted averaging have been translated to Bayesian frameworks (e.g., Kühl et al., 2002; Haslett et al., 2006; Holden et al., 2017). Pollen records contain information on the local climate during a time slice, where the spatial scale is constrained by the influx domain of horizontal pollen transport. Typically, macrofossils have a higher taxonomic resolution than pollen such that the climatic niche of the occurring taxa can be better constrained than with pollen alone (Bradley, 2015). Equilibrium simulations with earth system models (ESMs) produce a physically consistent estimate of the atmospheric and oceanic circulation and the regional energy balance given a set of forcings (boundary conditions). Important boundary conditions, for which information are available from proxy data and physical models, are insolation determined by the earth orbital parameters, greenhouse gas concentrations, ice sheet configurations, and land-sea masks. We use an ensemble of simulations from different ESMs to estimate a prior distribution, which contains a wide range of physically reasonable climate states. The combination of these two sources of information can be interpreted as a downscaling of forcing conditions via ESMs and an upscaling of local information contained in pollen records via spatial covariance matrices. The result is a spatially distributed and physically reasonable probabilistic climate reconstruction on continental domains.

We apply our framework to a mid-Holocene (MH, around 6ka) example for two reasons. First, compared with other time slices before the common era, the MH has a high proxy data coverage, particularly for Europe. Therefore, we can use raw pollen and macrofossil data with a sparse but relatively uniform spatial coverage over Europe as input for probabilistic transfer functions, while still having other reconstructions available, that can be compared with our results. Second, a multi-model ensemble of climate simulations with boundary conditions adjusted to the MH was produced in the Paleoclimate Modelling

Intercomparison Project Phase III (PMIP3) project (Braconnot et al., 2011). This ensemble is used to estimate the spatial prior distribution. The posterior distribution, which we estimate, is a multivariate probability distribution, with marginal distributions for each grid box, as well as spatial correlations and correlations between two climate variables, the mean temperature of the warmest month (MTWA) and the mean temperature of the coldest month (MTCO). For further analyses, we create samples

from this distribution, such that each sample is an equally probable estimate of the bivariate spatial field. In the context of temporal reconstructions these samples were called "climate histories" by Parnell et al. (2016). From the samples, quantitative properties of the climate state during the MH, which account for uncertainties, are computed. In addition, our framework can be used to compare the model-data mismatch of multiple ESMs, to analyse the consistency of a given proxy network, and to help in the identification of potential outliers.

This work is related to several concepts that were developed for applications in paleoclimatology. In recent years, several authors constructed Bayesian hierarchical models (BHMs) for paleoclimate reconstructions: Tingley and Huybers (2010) introduced a spatio-temporal BHM for reconstructions of the last millennium with an underlying structure that is stationary, linear, and Gaussian. Other authors developed temporal (Parnell et al., 2015) or small-scale spatio-temporal BHMs (Holmström et al., 2015). All of these approaches differ from our model in being purely proxy data driven. Additional information

on orbital configurations were incorporated by Gebhardt et al. (2008) and Simonis et al. (2012) via an advection-diffusion model which is combined with proxy data using a variational inference approach. Li et al. (2010) included information on solar, greenhouse gas, and volcanic forcing for spatially averaged reconstructions of the last millennium via linear regression. Annan and Hargreaves (2013) combined Paleoclimate Modelling Intercomparison Project Phase II (PMIP2) simulations with the syntheses of Bartlein et al. (2011) and MARGO Project Members (2009) in a multi-linear regression model. We build on

these approaches by incorporating fields that are simulated from a set of MH forcing conditions in a fully Bayesian framework. A different approach to combine proxy data and climate simulations for spatio-temporal reconstructions of the common era was developed by Steiger et al. (2014) and Dee et al. (2016) using so-called off-line data assimilation methods. They apply an ensemble Kalman filter, where the observation operators are forward models for proxy data, and the prior covariance is estimated from a database of transient climate simulations. Our purely spatial reconstructions can be interpreted as an off-line

data assimilation with only one time step. This reduced dimensionality permits the exploration of the full posterior distribution despite incorporating non-linear and non-Gaussian elements in the observation operator and the spatial interpolation scheme.

The structure of the paper is as follows. In Sect. 2, we describe the pollen synthesis and climate simulations which we use. This is followed by a detailed description of our proposed Bayesian framework in Sect. 3. Results from a comparison study of different ways to incorporate the climate simulations in the Bayesian framework and from our reconstruction of the European

MH climate are presented in Sect. 4. Finally, we discuss and summarize our methodology and results in Sect. 5 and 6.

## 2 Data

### 2.1 Proxy and calibration data

The pollen and macrofossil synthesis, that we use in this study, stems from Simonis et al. (2012) as part of the European Science Foundation project DECVeg (Dynamic European Climate-Vegetation impacts and interactions). Out of the four time slices (6ka, 8ka, 12ka, 13ka), which were compiled, we only use the 6ka dataset because there is no ensemble of climate simulations available for the other three time slices. For 50 paleosites, information on the occurrence of taxa is provided. 59 taxa occur at least at one site. For some sites, information from very nearby sites are combined into a joint sample. 15 of the sites combine macrofossil and pollen information, three samples contain just macrofossil data, and for 32 sites only pollen data is available. In general, the macrofossil data provides more detailed taxonomic information than pollen. Because pollen is more prevalent than macrofossil data, pollen samples are included at sites with macrofossils as well as from additional sites to provide a broader spatial picture of the European vegetation at the MH.

The 50 paleosites are sparsely but relatively uniformly distributed over Europe. Their locations are delimited by 6.5° W, 26.5° E, 37.5° N and 69.5° N. Compared with other recent syntheses like Bartlein et al. (2011), less records are included due to high quality control criteria. The raw pollen or macrofossil data and radiocarbon measurements, from which at least one was supposed to be close to 6ka, had to be available to recalculate age-depth models and ensure the use of calibrated radiocarbon dates as common time scale. Each site is assigned to the corresponding cell of a 2° by 2° grid which we use for our reconstructions. The locations of the proxy samples are depicted by black dots in Fig. 1. The full list of sites included in the synthesis can be found in Simonis et al. (2012). The list of taxa, which occur at the sites, is published in Simonis (2009).

Modern climate and vegetation data is used for the calibration of the transfer functions. The climate data is computed from the University of East Anglia Climatic Research Unit (CRU) 1961 to 1990 reference climatology (CRU TS v.4.01, Harris et al., 2014; Harris and Jones, 2017)). The vegetation data stems from digitized vegetation maps (Schölzel et al., 2002). The regions that are used for the transfer function calibration were determined individually for each taxa by pollen experts (Kühl et al., 2007). The number of calibration sites varies between 14.543 and 28.844, depending on the taxa.

### 2.2 Climate simulations

We use a multi-model ensemble of climate simulations which were run within PMIP3 with forcings adjusted to the MH. This includes changed orbital configurations and greenhouse gas concentrations (Braconnot et al., 2011). The ensemble contains all available MH simulations in the PMIP3 database (downloaded from the German Climate Computing Center (DKRZ) long term archive, available under https://cera-www.dkrz.de), which have a grid spacing of at least 2°. This constraint, which retains only the models with the smallest grid spacings, is chosen to better match the resolutions of proxy samples and simulations. The condition results in using seven model runs performed with the CCSM4, CNRM-CM5, CSIRO-Mk2-6-0, EC-Earth-2-2, HadGEM2-CC, MPI-ESM-P, and MRI-CGCM3. Properties of the included simulations are given in Table 1. The ensemble is a multi-model ensemble with common boundary conditions. The models are run to an equilibrium state (spin-up), followed by

typically around 100 simulated years to minimize noise due to high frequency internal variability. Therefore, the differences within the ensemble can be interpreted as modelling uncertainties (epistemic uncertainty).

The mean summer climate expressed as MTWA (Fig. 1a) from the MH ensemble is warmer than the CRU reference climatology (CRU TS v.4.01 over land, Harris et al. (2014), Harris and Jones (2017), and HadCRUT absolute over sea, Jones et al. (1999)) in large parts of Europe, especially eastern Europe and the Norwegian Sea. These areas coincide predominantly with areas of large ensemble spreads, expressed as point-wise empirical standard deviations in Fig. 1c. The standard deviations increase up to 4 K in some areas of southern and eastern Europe, which might originate from varying change patterns of the general circulation over Europe in the models. In contrast, the MH mean winter climate measured by MTCO in Fig. 1b shows a more dispersed structure with cooling in Fennoscandia, warming in the Mediterranean and Balkan peninsula, and mixed patterns in western and central Europe. The ensemble spread is predominantly small (Fig. 1d), but increases towards northern Europe with very large inter-model differences for the Norwegian Sea and eastern Fennoscandia.

## 2.3  Reconstruction variables

The spatial distribution of taxa is limited by three climatic factors: Temperature during growing season and in winter, and moisture availability (Huntley, 2012). Therefore, these three factors, translated into quantitative variables, are important for climate reconstructions from pollen. For large parts of Europe, it was shown in Simonis (2009) that the pollen and macrofossil synthesis is well-suited for joint reconstructions of July and January temperature as measures for the warmth of growing season and cold of winter, because at least one of these two variables is a limiting factor for most taxa growing in the mid and high latitudes of Eurasia during the Holocene. Instead, testing various climate variables as indicators for moisture availability was less promising since moisture availability is rarely a limiting factor for European taxa (Simonis, 2009). Hence, in this study, we choose MTWA and MTCO as the target variables for our climate reconstructions. This is a compromise between variables that are bioclimatically meaningful and variables for which accurate data is available to calibrate the transfer functions against modern vegetation and climate data. In the mid to high latitudes, MTWA and MTCO are highly correlated with July and January temperature, respectively, which is why they are also described as "functionally equivalent" (Bartlein et al., 2011).

To calculate MTWA and MTCO from time series of monthly averages, the data is first interpolated to the desired spatial grid. Then, for each hydrological year (October to September), the warmest and coldest month are extracted. We choose the hydrological instead of the calendar year to ensure that the months are taken from connected seasons. Afterwards, the climatological mean is calculated by averaging over the values for each year.

## 3  Methods

### 3.1  Bayesian framework

We use Bayesian statistics to combine a network of pollen samples with an ensemble of PMIP3 simulations because in this approach each source of information has an associated uncertainty that is naturally included in the inference process. In this

section, we specify the quantities that are combined in our reconstruction, and describe the inference algorithm that is used to create the results presented below.

In the following, we denote fossil pollen and macrofossil data by $P_p$, past climate by $C_p$, modern vegetation and climate data for the calibration of transfer functions by $P_m$ and $C_m$, respectively, and additional model parameters by $\Theta$. We are interested in the conditional distribution of $C_p$ and $\Theta$ given fossil pollen and macrofossil, modern vegetation, and modern climate data, i.e. we want to estimate the posterior distribution $\mathbb{P}(C_p, \Theta \,|\, P_p, C_m, P_m)$.

Applying Bayes' theorem to $\mathbb{P}(C_p, \Theta \,|\, P_p, C_m, P_m)$ (in the following, we omit normalizing constants), we get:

$$\underbrace{\mathbb{P}(C_p, \Theta \,|\, P_p, C_m, P_m)}_{\text{Posterior}} \quad \propto \quad \underbrace{\mathbb{P}(P_p, P_m \,|\, C_p, C_m, \Theta)}_{\text{Data Stage / Likelihood}} \quad \underbrace{\mathbb{P}(C_p, C_m \,|\, \Theta)}_{\text{Process Stage}} \quad \underbrace{\mathbb{P}(\Theta)}_{\text{Prior Stage}} . \tag{1}$$

Following Tingley and Huybers (2010), we call $\mathbb{P}(P_p, P_m \,|\, C_p, C_m, \Theta)$ the data stage, $\mathbb{P}(C_p, C_m \,|\, \Theta)$ the process stage, and $\mathbb{P}(\Theta)$ the prior stage. In paleoclimatology, the data stage is traditionally called transfer function, which in our case is formulated in a forward way. It probabilistically models the proxy data given climate variables and is described in detail in Sect. 3.2. The process stage stochastically interpolates the local climate information from the proxy data to a spatial domain and is described in Sect. 3.3. The prior stage defines prior distributions for the model parameters $\Theta$. These prior distributions are necessary to get a closed Bayesian model which ensures that the posterior is a valid probability distribution (Gelman et al., 2013).

To further structure the framework, we split the model parameters $\Theta$ into $\theta$, which are parameters associated with the data stage, and $\vartheta$, which are parameters that influence the process stage. We assume that $\theta$ and $\vartheta$ are a priori independent of each other and that the data stage is conditionally independent of $\vartheta$ given $C_p$. Furthermore, by construction, $P_m$ and $C_m$ only contribute to the reconstruction via the transfer function parameters, i.e. they are assumed to be independent of all other quantities. Hence, we can rewrite Eq. (1) and get

$$\mathbb{P}(C_p, \Theta \,|\, P_p, C_m, P_m) \quad \propto \quad \underbrace{\mathbb{P}(P_m \,|\, C_m, \theta)}_{\text{Calibration stage}} \quad \underbrace{\mathbb{P}(P_p \,|\, C_p, \theta)}_{\text{Observation stage}} \quad \mathbb{P}(C_p \,|\, \vartheta) \quad \mathbb{P}(\theta) \quad \mathbb{P}(\vartheta). \tag{2}$$

Here, $\mathbb{P}(P_m \,|\, C_m, \theta)$ is called calibration stage and $\mathbb{P}(P_p \,|\, C_p, \theta)$ is called observation stage (Parnell et al., 2015). The structure of the Bayesian model can be expressed by a directed acyclic graph as shown in Fig. 2. In the graph, each node represents a variable and the arrows indicate dependences of variables.

## 3.2 Transfer function

The Bayesian model uses probabilistic transfer functions to model proxy data, in our case occurrence information on taxa, given climate and transfer function parameters for which prior distributions have to be defined. From all the terms in Eq. (2), the calibration stage, the observation stage, and the prior distribution of the transfer function parameters $\mathbb{P}(\theta)$ are related to the transfer function. As described above, our main reconstruction target is the bivariate climate $C = (C_1, C_2)$, where $C_1$ is MTWA and $C_2$ is MTCO.

To reconstruct climate from the Simonis et al. (2012) synthesis, the probabilistic indicator taxa method (PITM) model is used, which is a well established transfer function to quantitatively constrain past climate states by occurrence information on

taxa extracted from pollen and macrofossil samples. It uses taxa which are sensitive to MTWA and MTCO and determines the climatic niche, where they occur, by fitting response functions. The classical indicator taxa method (Iversen, 1944) estimates binary limits, e.g. a taxa occurs above a certain temperature but not below it. PITM, also named pdf method in the literature (Kühl et al., 2002), is an extension of this method where probability distributions are fitted to acknowledge that most taxa have a preferred climate space but the transitions between climates where they usually occur and those where they do not grow is soft. Initially, Gaussian distributions were used for calibration (Kühl et al., 2002) against vegetation maps (Schölzel et al., 2002). Later, the model was extended to mixtures of Gaussians (Gebhardt et al., 2008) and quadratic logistic regression (Stolzenberger, 2011, 2017).

We integrate the forward formulation of PITM from Stolzenberger (2017) in our Bayesian framework. For each taxa, we fit a quadratic logistic regression model (response function) describing the probability of taxa occurrence for a given value of $C$. The idea of using quadratic logistic regression stems from the BIOMOD (BIOdiversity MODelling) software which is a model to predict species distributions (Thuiller, 2003). The regression for taxa $T$ contains linear and quadratic terms for each of the climate variables as well as an interaction term:

$$\mathbb{P}(T = 1 \,|\, C = (C_1, C_2)) \;=\; \mathrm{logit}\left(\beta_1^T + \beta_2^T C_1 + \beta_3^T C_2 + \beta_4^T C_1 C_2 + \beta_5^T C_1^2 + \beta_6^T C_2^2\right). \tag{3}$$

Here, logit denotes the logistic function and $\beta_1^T, ..., \beta_6^T$ are regression coefficients. This regression leads to a unimodal response function which is anisotropic but has two symmetry axes, as can be seen for dwarf birch (Betula nana) and European ivy (Hedera helix) in Fig. 3.

To fit response functions, vegetation data is used instead of modern pollen or macrofossil data, because it contains more accurate information on the occurrence of a taxa on the spatial scales of interest compared to modern pollen samples. Moreover, the nature of macrofossils makes it impossible to create a modern calibration dataset that contains pollen and macrofossils. The disadvantage of using vegetation data for the calibration is that the probability of presence of a taxa is only valid in vegetation space on the spatial scale taken for the training data but not in the pollen or macrofossil space, where an absence of a taxa in a pollen or macrofossil sample can have multiple non-climatic reasons like local plant competition or pollen transport effects, as well as local climatic effects below the resolution of our reconstruction such that the taxa did not grow in the immediate surrounding of the core location.

For the calibration against the modern dataset, we use presence (T=1) as well as absence (T=0) information on the taxa which can be justified by assuming that the vegetation maps contain accurate information on presence as well as absence of taxa. From the definition given in Sect. 2.3, it follows that at any location MTWA is larger or equal than MTCO. Formally incorporating this constraint in the inference leads to a non-linear condition on the regression parameters, which is very hard to implement. Therefore, we choose the more practical way of adding artificial absence information for combinations of MTWA and MTCO such that MTCO > MTWA. While this leads to transfer functions, which do not preclude reconstructions of MTCO values larger than MTWA, it is at least very improbable. To apply the response functions for individual taxa to a set of proxy data, we assume that proxy samples $P(s)$, where $s = 1, ..., S$ subscripts the proxy samples, are conditionally independent given a climate field and that, conditioned on $C(x_s)$, where $x_s$ is the location of the $s$-th sample, $P(s)$ is independent of the climate

at all other locations. This leads to the following probabilistic model for the set of modern vegetation samples

$$\mathbb{P}(P_m \,|\, C_m, \theta) = \prod_{s=1}^{S_m} \prod_{T \in T(P)} \mathbb{P}\left(P_m^T(s) \,|\, C_m(x_s), \beta_1^T, ..., \beta_6^T\right). \tag{4}$$

Here, $P_m^T(s)$ is the presence or absence of taxa $T$ in the $s$-th calibration sample, $T(P)$ is the set of all Taxa occurring in the fossil pollen and macrofossil synthesis, and $\theta := (\beta_i^T,\ i = 1, ..., 6,\ T \in T(P))$.

As described above, the absence of a taxa in a pollen or macrofossil sample can have reasons that are not included in the absence probability estimated from Eq. (4), as this calibration is only valid in the vegetation space. As information on the absence of a taxa in the vegetation space (i.e. absence of the taxa in the grid box of interest at the respective time slice) is not available from pollen and macrofossil data, the only reliable occurrence information of a taxa in the respective grid box in the past is the presence of the taxa in a pollen or macrofossil sample (Gebhardt et al., 2003). Hence, only occurring taxa are

included in the reconstruction step.

Violations of the assumption that taxa are treated as conditionally independent given climate, i.e. due to co-occurrence of taxa (Kühl et al., 2002), can lead to over-fitting and subsequently underestimation of uncertainty in the transfer functions. Therefore, a statistical preselection of taxa, which are present in a sample, is applied. This procedure uses the Mahalanobis distance (Mahalanobis, 1936) between the fitted distributions and is described in detail in Kühl et al. (2002) and Gebhardt et al.

(2008). For the pollen and macrofossil synthesis used in this study, the preselection was carried out by Simonis (2009) and we follow their results.

It is assumed that the transfer function acts locally, i.e. given the climate at location $x$, a pollen sample at $x$ is conditionally independent of the climate and proxy data at all other locations. Following these considerations, $\mathbb{P}(P_p \,|\, C_p, \theta)$ is given by

$$\mathbb{P}(P_p \,|\, C_p, \theta) := \prod_{s=1}^{S_p} \prod_{T \in T(s)} \mathbb{P}\left(P_p^T(s) \,|\, C_p(x_s), \beta_1^T, ..., \beta_6^T\right), \tag{5}$$

where $T(s)$ are the taxa occurring in sample $s$ and picked by the preselection procedure of Simonis (2009).

Finally, we define a prior distribution for $\theta$. We use a Gaussian distribution centred at $0$ and a marginal variance of 10 for each parameter $\beta_i^T$. Due to the absence of prior information on the correlation structure, we assume independence between the taxa as well as within a taxa. Due to the high information content in the calibration data set, the influence of the prior on the parameter estimates is negligible. Using a flat prior for $C_p(x_s)$ and removing spatial correlations, local climate reconstructions

at the locations of the proxy samples can be calculated. These reconstructions depend only on the proxy data in grid box $x_s$. Results of local MH reconstructions for each grid box with proxy data are shown in Fig. 4, where the local reconstruction means and the marginal 90% credible intervals (CIs) are plotted for MTWA and MTCO.

Local reconstructions can also be used to evaluate the ability of the transfer functions to reconstruct modern climate which provides a reference for possible regional biases. For the PITM model such evaluations have been performed by Gebhardt et al.

(2008) and Stolzenberger (2011). Both evaluations show that the model tends to underestimate north-south gradients leading to positive biases in Fennoscandia, and slightly negative biases in the Mediterranean. The biases as well as the uncertainties are

larger for winter temperature than for summer. Therefore, results for MTCO in northern Fennoscandia should be treated with caution, while for all other regions biases of the reconstruction means are within reconstruction uncertainties.

A disadvantage of the PITM version used in this study, is the inconsistent use of calibration and fossil data by using presence and absence information on taxa for the calibration but only occurring taxa in the reconstruction. Despite this inconsistency, the reconstructions in this study are in agreement with previous versions of PITM, where this inconsistency with the calibration did not appear as previously only occurrence information were used to fit the probability density functions. However, there is no simple solution for the problem that the calibration is in vegetation space whereas the absence of taxa in the fossil samples is an information in the pollen or macrofossil space. A promising idea might be to model the absence due to non-climatic reasons as zero-inflation by adding a latent variable to estimate the detection probability of a taxa (MacKenzie et al., 2002). But the estimation of detection probabilities is a very challenging task because it depends on many factors like pollen influx area, local topography, soil properties, and plant competition which might change over time. It is a priori unclear which of these factor can be marginalized and which have to be included as covariates. In addition, the Simonis et al. (2012) synthesis combines macrofossils with pollen data. The processes that influence the detection probability of macrofossils are very different than for pollen. Therefore, a different detection probability has to be estimated for pollen than for macrofossils. Resolving the described issues is an interesting direction for future research, which requires extensive cooperation of (paleo)climatologists, (paleo)botanists, and statisticians, but is beyond the scope of this study.

### 3.3 Process stage

Similar to data assimilation approaches in numerical weather and climate prediction, the ensemble of climate simulations is used to control the spatial structures of the reconstruction and to constrain the range of physically possible climate states for a given external forcing by computing a spatial prior distribution from the ensemble members. This distribution is combined with interpolation parameters $\vartheta$ to facilitate a more flexible adjustment to the proxy data in the process stage of Eq. (2). The estimation of the prior distribution is hampered by the small number of ensemble members $K = 7$.

It is not obvious which method for estimating the prior distribution is best suited for the problem on hand and which additional model parameters are appropriate to preserve as much physical consistency contained in the climate simulations as possible but to correct for climate model inadequacies. Therefore, we perform a comparison study of six process stage models, that are composed of three techniques to formulate the process stage and two choices for the involved spatial covariance matrix. All those models are inspired by methods used in data assimilation, postprocessing of forecasts, and climate change detection and attribution. The use of climate simulations in the process stage allows not just interpolation between proxy samples but also structural extrapolation through the eigenvectors of the spatial covariance matrix and the process stage parameters.

### 3.3.1 Gaussian model

The most common approach in the data assimilation literature is to assume that the ensemble members are independent and identically distributed (iid) samples from an unknown Gaussian distribution of possible climate states (Carrassi et al., 2018). In the following, the climatological means of the $K$ ensemble members are denoted by $\mu_k$. Subsequently the spatial prior

distribution is given by $\mathcal{N}(\bar{\mu}, \Sigma_{\text{prior}})$, where $\mathcal{N}$ denotes a Gaussian distribution, $\bar{\mu}$ is the ensemble mean, and $\Sigma_{\text{prior}}$ is a spatial covariance matrix, which is given by a regularized version of the empirical covariance

$$\Sigma_{\text{emp}} = \frac{1}{K-1}\sum_{k=1}^{K}(\mu_k - \bar{\mu})(\mu_k - \bar{\mu})^t. \tag{6}$$

The superscript $t$ denotes the matrix transpose. Hence, the covariance matrix is based on the inter-model differences as an estimate of epistemic uncertainties. The regularization techniques of $\Sigma_{\text{emp}}$ are specified below. The Gaussian distribution is multivariate and the state vector has dimension $N$, where $N$ is the number of grid boxes times the number of jointly reconstructed variables.

The main advantage of this Gaussian model (GM) is that inference becomes simpler than in more complex probability density estimation techniques because the prior distribution is unimodal and Gaussian. The disadvantage is that it relies on the strong assumption that the samples $\mu_k$ are iid samples from an unknown Gaussian distribution. This assumption tends to be more realistic for samples from just one ESM, whereas statistics of multi-model ensembles are often not well described by purely Gaussian distributions (Knutti et al., 2010). A second disadvantage of this model is that the absence of additional parameters limits the possibilities to adjust the posterior distribution to the proxy data and correct for climate simulation inadequacies. The third disadvantage of the GM is that spatial structures of the individual models, which are directly derived from the physical equations solved in the ESMs, get lost by averaging over all ensemble members. Nevertheless, in many climate prediction applications multi-model averages outperformed each individual ESM (Krishnamurti et al., 1999, e.g.).

### 3.3.2 Regression model

A relaxation of the assumptions of the GM is the second model, that we call the regression model (RM) because it is inspired by regression based models popular in postprocessing and climate change detection and attribution (Hegerl and Zwiers, 2011). In the RM, the iid assumption is dropped for the first moments of the process stage by introducing weighted averages of the ensemble members with variable weights $\lambda_k, k = 1, ..., K$. This means, that samples, which fit better to the proxy data, are weighted higher in the posterior. The sum of the weights is set to one such that unrealistically warm or cold state are prevented. This leads to the process stage model

$$\mathbb{P}(C_p|\lambda_1,...,\lambda_K) = \mathcal{N}\left(C_p \,\Big|\, \sum_{k=1}^{K}\lambda_k\mu_k, \Sigma_{\text{prior}}\right), \tag{7}$$

and an additional prior distribution for the model weights

$$\mathbb{P}(\lambda) = \text{Dir}\left(\lambda_1,...,\lambda_K \,\Big|\, \tfrac{1}{2},...,\tfrac{1}{2}\right). \tag{8}$$

Dir denotes a Dirichlet distribution, which is chosen as prior distribution because is guarantees that the weights take values between zero and one and sum up to one. Conditioned on the ensemble member weights, the process stage distribution is Gaussian, but non-Gaussianity is permitted through the variable weights.

In addition, the RM has the advantage of possessing more degrees of freedom compared to the GM. The inference process becomes a little more involving than for the GM because the ensemble member weights have to be estimated, too, but the conditional Gaussian distribution of $C_p$ helps designing efficient inference algorithms. Similar to the GM, spatial structures of the individual models can get lost, as we average over different ESM climatologies.

### 3.3.3 Kernel model

The third model has been introduced in the data assimilation literature by Anderson and Anderson (1999) to combine particle and Gaussian filtering approaches, where the particle part helps capturing non-Gaussian and non-linear features but the efficiency of Gaussian approximations in high dimensional filtering situations is still exploited. This kernel model (KM) assumes that each ensemble member is a sample from an unknown distribution of possible climate states given a set of forcings, but it does not assume that this unknown distribution is Gaussian. Instead, non-parametric kernel density estimation techniques (Silverman, 1986), where the probability distribution is given by a mixture of multivariate Gaussian kernels, are used. Each ensemble member climatology corresponds to the mean of a kernel.

Ideally, the covariance matrix of each kernel would correspond to the respective ESM, such that the spatial autocorrelation of that ESM is preserved when we sample from its kernel. Unfortunately, there is only one MH run available for each ESM and the internal variability in those runs is much smaller than the inter-model differences. Using the internal variability of those runs would thus lead to very distinct kernels and allow too few climate states. Therefore, the covariance of each kernel is estimated from the inter-model differences as a compromise that allows to sample from a much broader range of states even though autocorrelation of the individual models is lost. This compromise is a very common choice in kernel based probability density approximations (Liu et al., 2016; Silverman, 1986, Chapter 3 and 4) if there is no good estimate for the covariance corresponding to each kernel available.

Compared to the GM, the empirical covariance matrix $\Sigma_{\text{emp}}$ is scaled by the Silverman factor (Silverman, 1986)

$$f := \left( \frac{4}{K \cdot (N+2)} \right)^{\frac{2}{N+4}},$$
(9)

which optimizes the variances of the kernels, Hence, in the KM the scaled empirical covariance matrix $\tilde{\Sigma}_{\text{emp}}$, given by $f \cdot \tilde{\Sigma}_{\text{prior}}$, is regularized leading to the spatial covariance matrix $\tilde{\Sigma}_{\text{prior}}$. Note that the small number of ensemble members compared to the dimension of the probability distribution leads to a standard deviation reduction of only around 2% in our applications.

Each kernel gets an assigned weight $\omega_k$, $k = 1, ..., K$, which is inferred in the Bayesian framework. The weights sum up to one. The resulting process stage is a mixture distribution

$$\mathbb{P}(C_p | \omega_1, ..., \omega_K) = \sum_{k=1}^{K} \omega_k \, \mathcal{N}\left( C_p | \mu_k, \tilde{\Sigma}_{\text{prior}} \right).$$
(10)

A Dirichlet distributed prior is used for $\omega$ with parameter $\frac{1}{2}$ for each of the $K$ components. A computational disadvantage of the KM is that the process stage is multi-modal and non-Gaussian. We augment the model by an additional parameter $z$, which follows a categorical distribution, denoted by Cat, to restore that $C_p$ is Gaussian conditioned on the $\omega$ and $z$. $z$ selects a kernel

$k$ according to its weight $\omega_k$, i.e. $z$ is defined such that

$$\mathbb{P}(\omega) = \text{Dir}\left(\omega_1, ..., \omega_K \mid \tfrac{1}{2}, ..., \tfrac{1}{2}\right) \tag{11}$$

$$\mathbb{P}(z \mid \omega) = \text{Cat}(z_1, ..., z_K \mid \omega_1, ..., \omega_K) \tag{12}$$

$$\mathbb{P}(C_p \mid z) = \prod_{k=1}^{K} \left(\mathcal{N}(C_p \mid \mu_k, \Sigma_{\text{prior}})\right)^{z_k}. \tag{13}$$

Integrating out $z$ yields the mixture distribution Eq. (10).

Two advantages of the KM are that it is not assumed that the unknown prior distribution is Gaussian and that the kernels do not rely on an iid assumption for their first moment properties. On the other hand, the KM relies on an iid assumption for the second moment properties, which is the compromise described above due to the absence of a suitable estimate of the uncertainty structures corresponding to each ensemble member. The KM preserves the spatial structures of each ESM in

the first moments of the kernels and when sampling from Eq. (13), the mean of the sample belongs to one ESM and is not a weighted average over all ensemble members. This preservation of physical consistency reduces the degrees of freedom compared to the RM. For example, when the true climate state lies exactly between $\mu_1$ and $\mu_2$ and far away from all other ensemble members, the weights of $\mu_1$ and $\mu_2$ can be increased compared to the other members in the KM, but the mode cannot be changed to $\frac{1}{2}(\mu_1 + \mu_2)$, which is possible in the RM. Another disadvantage of the KM is that the multi-modality makes the

design of efficient inference algorithms a lot more challenging.

### 3.3.4  Glasso based covariance matrices

The first technique to regularize the empirical covariance matrix (the scaled empirical covariance in the KM), which is applied in this study, is the graphical lasso algorithm (glasso, Friedman et al., 2008, implemented in the R-package glasso). This algorithm approximates the precision matrix (inverse covariance) by a positive definite, symmetric, and sparse matrix $\Sigma_{\text{prior}}^{-1}$.

Therefore, $\Sigma_{\text{prior}}$ is a valid $N$-dimensional covariance matrix. Glasso maximizes the penalized log-likelihood

$$\log \det \Sigma_{\text{prior}}^{-1} - \text{trace}(\Sigma_{\text{emp}} \Sigma_{\text{prior}}^{-1}) - \rho \, \|\Sigma_{\text{prior}}^{-1}\|_1, \tag{14}$$

where $\rho$ is the penalty parameter, $\|\cdot\|_1$ is the vector $L_1$-norm, and the first two terms are the Gaussian log-likelihood. Because applying the glasso algorithm is computationally expensive, it is not feasible to formally include $\rho$ in the Bayesian framework. Instead a suitable value of $\rho$ has to be determined prior to the inference. In this study, $\rho$ is chosen such that $\Sigma_{\text{prior}}$ is a numerically

stable covariance matrix and the performance in cross-validation experiments (CVEs) is optimized. Technical details of the determination of $\rho$ are described in Appendix A.

The advantage of the glasso approach is that the empirical matrix can be approximated very closely and the sparseness of the precision matrix facilitates the use of efficient Gaussian Markov random field (GMRF) techniques (Rue and Held, 2005) in the inference algorithm. A disadvantage is that no new spatial structures are added to $\Sigma_{\text{emp}}$. Therefore, the effective number of

spatial modes is much smaller than the dimension of the climate vector, which can lead to a collapse onto a very small subspace of the $N$-dimensional state space and subsequently biases and under-dispersion.

### 3.3.5   Shrinkage based covariance matrices

To overcome the deficiencies of the glasso approach, we propose an alternative covariance regularization technique, which combines the empirical correlation matrix of the climate simulation ensemble with a regular correlation matrix. A so-called shrinkage approach (Hannart and Naveau, 2014) is used, which is a weighted average of the empirical correlation matrix and a reference matrix, which in our case contains additional spatial modes such that the effective number of spatial modes in the covariance matrix is increased. This allows deviations from the spatial structures prescribed by the climate simulation ensemble and is therefore a strategy to account for climate model inadequacies.

Let $\Psi_{\mathrm{emp}}$ be the empirical correlation matrix of the climate simulation ensemble, which is related to $\Sigma_{\mathrm{emp}}$ by

$$\Sigma_{\mathrm{emp}} = \mathrm{Diag}(\Sigma_{\mathrm{emp}})^{\frac{1}{2}} \, \Psi_{\mathrm{emp}} \, \mathrm{Diag}(\Sigma_{\mathrm{emp}})^{\frac{1}{2}}, \tag{15}$$

where $\mathrm{Diag}(\Sigma_{\mathrm{emp}})$ denotes a diagonal matrix with the the same diagonal entries as $\Sigma_{\mathrm{emp}}$, and the exponent $\frac{1}{2}$ means that the square root of each diagonal entry is taken. Replacing $\Psi_{\mathrm{emp}}$ by a weighted average of $\Psi_{\mathrm{emp}}$ and a shrinkage target $\Phi$ leads to the shrinkage covariance matrix

$$\Sigma_{\mathrm{prior}} = \mathrm{Diag}(\Sigma_{\mathrm{emp}})^{\frac{1}{2}} \, (\alpha \, \Psi_{\mathrm{emp}} + (1-\alpha) \, \Phi) \, \mathrm{Diag}(\Sigma_{\mathrm{emp}})^{\frac{1}{2}}. \tag{16}$$

$\alpha$ is the weighting parameter, which takes values between zero and one. $\Phi$ is computed from a numerically efficient GMRF approximation of a stationary Matérn correlation matrix (Lindgren et al., 2011). Matérn correlation functions correspond to diffusive transport of spatial white noise forcing. The Matérn correlation matrix is controlled by three parameters, the smoothness, the range $\rho$, and the anisotropy $\nu$. We fix the smoothness at a value which corresponds to the application of a standard Laplace operator. $\rho$ controls the decorrelation length, and $\nu$ parameterizes the length of the meridional compared to the zonal decorrelation length. For joint reconstructions of multiple climate variables, independent correlation matrices for each variable are combined in a block structure. Details about the definition of $\Phi$ are given in Appendix B.

Ideally, the parameters $\alpha$, $\rho$, and $\nu$ are estimated from the proxy data. But initial tests with weakly informative priors showed that the parameters cannot be constrained by the available proxy data, because the signal in the proxy data is not informative enough to infer second moment properties of the process stage. Therefore, an ensemble of parameter combinations is created from fitting the shrinkage model to each of the climate simulation ensemble members given all other members. This results in seven consistent sets of $\alpha$, $\rho$, and $\nu$. Those are passed to the reconstruction framework, such that each combination is chosen with a probability inferred from the proxy data. Thereby, each parameter combination is based on a fit against physically consistent structures, and uncertainty in the parameters is included in the inference, but the problem of non-identifiability of the parameters from proxy data alone is reduced. The parameter estimates depend strongly on the chosen ESM, which leads to combinations that cover a wide range of possible values.

The main advantage of the shrinkage approach over the glasso based matrices is that more spatial modes are included in the covariance matrix. Thereby, the collapsing of the reconstruction towards a very low-dimensional subspace is mitigated and stronger deviations of the reconstruction from the climate simulation ensemble are facilitated. A disadvantage compared to the

glasso estimated covariance matrices is that neither the shrinkage covariance matrix $\Sigma_{\text{prior}}$ nor its inverse are sparse matrices such that the numerical inference algorithms are more costly.

## 3.4 Inference strategy

Because PITM is non-Gaussian and non-linear, the posterior climate does not belong to a standard probability distribution. Therefore, Markov chain Monte Carlo (MCMC) techniques are used to asymptotically sample from the correct posterior distribution. These samples allow analyses beyond summary statistics like means and standard deviations. A Metropolis-within-Gibbs strategy is implemented, which means that in each update of the Markov chain, we sample sequentially from the full conditional distributions (i.e. the distribution of the respective variable given all other variables) of $\theta$, $\vartheta$, and $C_p$. This strategy is chosen because for many variables the full conditional distributions follow probability distributions for which efficient sampling algorithms exist, and for the remaining variables Metropolis-Hastings updates are used for sampling.

To sample the regression parameters $\theta$ in Eq. (3) to (5) efficiently, the data augmentation scheme of Polson et al. (2013) is used. For taxa $T$, the full conditional is only depending on $C_p$, $C_m$, $P_p^T$, and $P_m^T$, but not on other taxa. Therefore, we can sample $\beta_1^T, ..., \beta_6^T$ independently from the other taxa. Polson et al. (2013) introduce help variables $\gamma_l^T, l = 1, ..., L$, where $L$ is the number of observations of taxa $T$, such that $\mathbb{P}(\gamma_l^T \,|\, \beta_1^T, ..., \beta_6^T, C_m, C_p)$ is Pólya-Gamma (PG) distributed, and $\mathbb{P}(\beta_1^T, ..., \beta_6^T \,|\, P_m^T, P_p^T, \gamma_1^T, ..., \gamma_L^T)$ is Gaussian. Therefore, the MCMC algorithm samples alternately from a PG distribution using the sampler of Windle et al. (2014) and from a Gaussian distribution. The PG sampler is implemented in the R package BayesLogit (Windle et al., 2013).

To sample from the full conditional of $C_p$, we separate the grid boxes $x_P$ with at least one proxy record from those without any proxy records denoted by $x_Q$. There is no closed form available for the full conditionals of $C_p(x_P)$. Therefore, we use a random walk Metropolis-Hastings algorithm to update $C_p(x_P)$ sequentially for all members of $x_P$. As the transfer functions act locally, $C_p(x_Q)$ is conditionally independent of $P_p$ given $C_p(x_P)$ and $\vartheta$. Therefore, we subsequently update $C_p(x_Q)$ by sampling from $\mathbb{P}(C_p(x_Q)\,|\,C_p(x_P), \vartheta)$ which is Gaussian.

Sampling from $\vartheta$ depends on the particular process stage model. In models with shrinkage covariance matrix, $\alpha$, $\rho$, and $\nu$ are sampled from the $K$ parameter combinations in a Metropolis-Hastings step. The weights $\lambda$ in the RM are sampled from a random walk type Metropolis-Hastings update. In the KM, Eq. (11) to (13) lead to full conditionals for $\omega$ and $z$, which are again Dirichlet and categorically distributed but with updated parameters. Therefore, Gibbs sampling can be used to update $\omega$ and $z$.

The multi-modality of the KM makes inference for this model a lot more challenging than for the GM and RM. The problem of efficient MCMC algorithms for multi-modal posterior distributions is a widely acknowledged issue in the literature (Tawn and Roberts, 2018) and in this study Metropolis coupled Markov chain Monte Carlo (MC[3]; Geyer, 1991), which is also known as parallel tempering, is used to overcome this issue. Details of this procedure are provided in Appendix D.

To speed up the inference, grid boxes with proxy data and those without proxy data are treated sequentially. Because of the conditional Gaussian structure of the process stage and because the grid boxes without proxy data are not influencing the posterior of $\vartheta$, $C_p(x_Q)$ is integrated out to get an estimate of the joint distribution of $\Theta$ and $C_p(x_P)$. In a second step,

we sample from $C_p(x_Q)$ conditioned on $C_p(x_P)$ and $\Theta$, which leads to joint samples of $C_p$ from the asymptotically correct posterior distribution. This strategy reduces the computation time of each MCMC update due to faster matrix operations.

The remaining bottleneck in computation time is the estimation of the transfer function parameters due to the large modern calibration set. While in theory the observation layer influences the updates of $\theta$, in practice the influence of Eq. (5) on the posterior of $\theta$ is negligible. Therefore, a modularization approach (Liu et al., 2009) is used similar to Parnell et al. (2015) in CVEs, where a sequence of reconstructions with slightly changed proxy networks is computed. This means that feedback between Eq. (4) and Eq. (5) is cut by first drawing as many MCMC samples as necessary from $\theta$ using only Eq. (4). Thereafter, $C_p$ is reconstructed using these samples instead of sampling $\theta$ from its full conditional.

Detailed formulas for the full conditional distributions are given in Appendix C. Pseudo-code for the MCMC and MC[3] algorithms is provided in the Supplement. For a 798 dimensional climate posterior as it is the case in joint reconstructions of MTWA and MTCO, and 45 grid boxes that contain at least one proxy record, 75,000 MCMC samples are created. The first 25,000 samples are discarded as burn-in. To reduce the autocorrelation of subsequent samples, every fifth sample is extracted to create a set of 10,000 posterior samples which is used for further analyses. On a standard desktop computer, reconstructions with the modularized model can be computed in approximately 30 minutes. The convergence of all MCMC variables is checked using the Gelman-Rubin-Brooks criterion (Brooks and Gelman, 1998) implemented in the R package coda (Plummer et al., 2006).

## 4    Results

In this section, results from a comparison study of the six different process stage models are presented. Then, the MH reconstruction for Europe with the Simonis et al. (2012) synthesis and the PMIP3 MH ensemble is presented. We study the mean and uncertainty structure of the posterior distribution, the fit of different ensemble members to the proxy data, the added value of the reconstruction, and the results of joint compared to separate MTWA and MTCO reconstructions.

### 4.1    Comparison of different process stage frameworks

In this section, the reconstruction skill of the three process stage formulations (GM, RM, KM) and the two covariance models (glasso, shrinkage) are compared using two types of experiments. Identical twin experiments (ITEs) use the climate simulation ensemble by simulating pseudo-proxy data from one ESM and trying to reconstruct that reference climatology from the simulated proxies and the remaining ensemble members. These experiments facilitate the understanding of different modelling approaches for the process stage in a controlled environment. In particular, the ability of the Bayesian frameworks to reconstruct the climate can be evaluated without having to rely on indirect observations as it is the case in real paleoclimate applications where the true climate state is unknown. The second type of experiments are CVEs, where spatial reconstructions with the Simonis et al. (2012) synthesis are performed but the samples from one grid box are left out. Then, the reconstructions for this grid box are evaluated against the left-out sample in the vegetation space by applying the PITM forward model to the reconstruction. The advantage of these experiments is that the models are compared in a real-world setting. The disadvantage of

CVEs is that the goal of a reconstruction is to reconstruct climate but there are no direct observations of paleoclimate available such that evaluations against observations have to be indirect. Evaluating prediction models against indirect observations in the observation space through forward modelling is also a recent way of model skill evaluation in weather forecasting.

### 4.1.1 Identical twin experiments

ITEs make use of the fact that the PMIP3 ensemble is a multi-model ensemble such that the models can produce fairly different climatologies. Therefore, trying to reconstruct the climate state of one ESM given the others is a more realistic test environment than doing the same with single-model ensembles. The first step in an ITE is to choose a reference climate model with climate state $C_p^{\text{true}}$. Then, for each grid box, that contains samples from the Simonis et al. (2012) synthesis (denoted by $x_P$ in accordance with the notation in Sect. 3.4), pseudo-proxies are simulated. The proxies are simulated according to a Gaussian approximation

of the uncertainty structure of the local reconstructions depicted in Fig. 4, which means that it is described by a bivariate covariance matrix $\Sigma_p^{x_s}$ for each grid box $x_s$ with proxy data. The pseudo-proxies are assumed to be unbiased with a bivariate Gaussian distribution, i.e.

$$P(x_s) \sim \mathcal{N}\left(C_p^{\text{true}}(x_s), \Sigma_p^{x_s}\right), \quad x_s \in x_P. \tag{17}$$

Using unbiased Gaussian pseudo-proxies is a common strategy to test climate field reconstruction techniques (e.g. Gomez-

Navarro et al., 2015). It allows a direct study of the ability of the process stage methods to estimate spatial climate fields from sparse and noisy proxy data, without having to factor in potential biases in the transfer function. With the simulated proxies, the Bayesian framework is applied to compute a probabilistic spatial reconstruction, but the reference model climatology is removed from the climate model ensemble. ITEs are performed with each of the seven PMIP3 ensemble members as reference climate state, and the six different process stage configurations. For each of those 42 combinations, five randomized ITEs are

computed to separate random from systematic issues.

The evaluation of the ITEs focuses on biases in the reconstructions, potential under-dispersion which is a typical phenomenon in data assimilation applications, and, as a combination of those two issues, the ability of the reconstruction to probabilistically predict past climate.

Averaged over all ITEs with the same process stage model and averaged in space, the mean deviation between reference

climate and posterior mean as a measure for systematic biases is close to 0 K for all process stage models with values between -0.14 K for the shrinkage KM and +0.03 K for the shrinkage RM, but the variation across ITEs is larger for the glasso covariance models (standard deviation around 0.31 K) than for the shrinkage covariance models (standard deviation around 0.24 K). This shows that the additional spatial modes in the shrinkage covariances make the reconstructions more robust (Table 2, Fig. 5a). The standard deviations for MTCO reconstructions tend to be larger than for MTWA which can be explained by

the larger noise level in the local MTCO reconstructions. This makes the spatial MTCO reconstructions more susceptible to biases. Concerning the spatial patterns of mean deviations, all ITEs with glasso covariance matrices have larger local biases than those with shrinkage matrices (see additional figures in Supplement). While the magnitude of biases for the GM and RM models with shrinkage covariances is much smaller, the magnitude of local deviations of the shrinkage KM model is just

slightly smaller than for glasso covariances. This shows that the models with shrinkage matrix can reconstruct spatial patterns better than the models that use glasso due to more degrees of freedom in the covariance matrix. In addition, averaging over different climate models in the mean of the process stage seems to be a more effective strategy as the GM and RM reproduce the spatial structures better than the KM.

The higher number of spatial modes in the shrinkage covariances leads to larger uncertainty estimates in the posterior distribution than for the glasso models, because the limited information contained in the proxy data can constrain only a small number of spatial modes (Table 2). To study dispersiveness of the reconstruction coverage frequencies for 50% and 90% CIs are calculated. This means that the frequency of the reference climate state to be included in the respective CIs is computed. For the 50% CIs, coverage frequencies below 50% indicate under-dispersiveness, whereas values above 50% indicate over-dispersion.

Similarly, the target for the 90% CIs is 90%. In all ITEs, the glasso models are under-dispersive, and the shrinkage models are over-dispersive (Table 2, Fig. 5c). The coverage frequency for 50% CIs is below 41% in all ITEs with glasso covariance matrix (mean close to 30% in all three experiments) and above 56% in all ITEs with shrinkage covariance matrix (mean around 78%). Similarly, the coverage frequencies for 90% CIs are below 77% for all glasso ITEs and above 94% for all ITEs with shrinkage matrix (Fig. 5d).

At most grid boxes of the ITEs with glasso based covariance matrix, the coverage frequencies are below the target values, whereas they are above the desired values at almost all grid boxes in the ITEs with shrinkage matrix (see additional figures in Supplement). The values are closest to the target near grid boxes with proxy data in the glasso as well as the shrinkage matrix ITEs. This effect is more pronounced for the 50% coverage frequencies than the 90% coverage frequencies, which indicates that the reconstruction identifies the centers of the probability distribution better than its tails, which is not surprising

considering the simplicity of all the process stage models and the small ensemble size. In particular, the process stage models do not contain parameters which control the tail behaviour.

    To analyse the combined effect of biases and dispersiveness, the continuous ranked probability score (CRPS) is computed. This is a common strictly proper score function for evaluating probabilistic predictions (Gneiting and Raftery, 2007), in our case the ability of a reconstruction method to probabilistically predict past climate from sparse and noisy data. It is a generalization

of the absolute error to probabilistic forecasts (Matheson and Winkler, 1976), given by

$$\text{CRPS}\left(F, C_p^{\text{true}}(x)\right) := \int_{-\infty}^{\infty} \left(F(y) - \delta_{(y \geq C_p^{\text{true}}(x))}\right)^2 dy, \tag{18}$$

where $F$ is the cumulative distribution function of the probabilistic reconstruction $C_p$ at grid box $x$, and $C_p^{\text{true}}(x)$ is the reference climate state at $x$. Defined in that way, the CRPS has a unique minimum at 0 and is positive unless $F$ is a perfect prediction.

    With a spatially averaged mean around 1 K for all three models, the ITEs with glasso covariance matrices have a higher

CRPS than the ITEs with shrinkage matrices that have a spatially averaged mean around 0.4 K (Table 2, Fig. 5b). This is a result of larger biases on the grid box level and under-dispersiveness of the posterior distribution. In addition, the variability between the ITEs with the same process stage model is higher for models with glasso covariance, which shows that these models are less robust. The MTWA CRPS is slightly lower than the MTCO CRPS since the local reconstructions constrain MTWA more

than MTCO. Among the process stage models with shrinkage matrix, the KM performs slightly worse than the GM and the RM which is a result of the larger biases on the grid box level described above. The spatial structures of CRPS reflect the mean deviation patterns (Fig. 6). This is an effect of more pronounced spatial patterns in the mean deviations compared to dispersiveness.

### 4.1.2 Cross-validation experiments

CVEs are a way to understand the ability of a spatial reconstruction method to produce consistent estimates. In paleoclimatology, the issue is that all observations are indirect, which means that negative evaluations can result from errors in the process stage or the data stage. For example, even if the climate reconstruction at grid box $x$ is accurate, the evaluation against the left-out data can be negative if the transfer function that translates data $C_p(x)$ into $P_p(x)$ is biased or its uncertainty estimate is misspecified. Hence, the assumption behind CVEs is that the data stage is unbiased or at least consistently biased among different proxy samples. Cross-validations are evaluated in the observation space. In this study, this is the vegetation space, i.e. the occurrence of taxa in a grid box. As the only reliable information that are available from the pollen and macrofossil synthesis on the vegetation composition in a grid box is the presence of certain taxa, this is also the only data that is used for the evaluation. Due to the sparseness of the proxy network, leave-one-out CVEs are performed and not more data is left out in each experiment.

In each CVE, a reconstruction with the Bayesian framework is computed with all proxy samples except for those in one grid box $x$. Then, the reconstruction $C_p(x)$ at grid box $x$ is extracted and treated as probabilistic prediction of the climate at $x$. Next, the PITM forward model is applied to $C(x)$ for each sample $P_p$ located in grid box $x$ to produce probabilistic predictions of the occurrence of the taxa that are found in those samples. This prediction of the occurrence of a taxa $T$ is represented by the probability of presence $p \in [0,1]$. A common score function for binary variables is the Brier score (BS; Brier, 1950) given by

$$BS(T) := \tfrac{1}{2}\left((\delta_T(1) - p)^2 + (\delta_T(0) - (1-p))^2\right) = \begin{cases} 1 + p^2 - 2p & \text{if } T = 1 \\ p^2 & \text{if } T = 0 \end{cases} \tag{19}$$

where $\delta_T$ denotes the indicator function of taxa $T$. The BS takes values between zero and one, where zero corresponds to a perfect prediction and one to the worst possible prediction. Stolzenberger (2017) used the BS to assess the skill of PMIP3 simulations for the MH using a network of pollen and macrofossil records. The PITM forward model is applied to each MCMC sample, which leads to a set of probabilistic predictions $p_j(T)$, $j = 1, ..., J$ for taxa $T$. Predictions are calculated for each taxa which occurs in sample $P(s)$. The joint score of $P(s)$ is then calculated by averaging the BS of each taxa and prediction:

$$BS(P(s)) := \frac{1}{|T(s)|J} \sum_{T \in T(s)} \sum_{j=1}^{J} \left(1 + p_j(T)^2 - 2p_j(T)\right). \tag{20}$$

If multiple samples are assigned to one grid box, the mean score of those samples is taken.

A problematic step in the methodology described above is that the BS is only evaluated for occurring taxa and not for those which are absent in the sample. This can make the BS improper when comparing statistical models that predict the presence

or absence of taxa. However, the goal of the methodology described above is an indirect evaluation of predictions of past climate via transfer functions. In that context, it would lead to inconsistencies between the local reconstructions and the BS evaluations if taxa that are absent in the proxy synthesis were included as there is currently no model available to accurately estimate detection probabilities of pollen and macrofossil data (see the discussion in Sect. 3.2). Circumventing this issue is

beyond the scope of this study. It should be noted that for each taxa the BS are a convex function of climate and minimal for a unique climate state. These two properties make the methodology described above useful for the comparison of climate reconstructions and the indirect evaluation of climate field reconstruction methods.

The models with glasso covariances perform slightly worse than those with shrinkage covariances, as the mean BS takes values of 0.186 (GM, RM ) or 0.187 (KM) for the glasso based models compared to values between 0.161 and 0.165 for models

with shrinkage covariances (Table 2). Similar to the ITEs, the differences between models with different covariance types are larger than those with the same covariance model. Accordingly, the largest differences at individual gridboxes between models with different covariance matrix types are on the order of $10^{-1}$, but only on the order of $10^{-2}$ between models with the same covariance type. The models with shrinkage covariance matrices tend to perform better in western Europe and Fennoscandia, whereas in central and eastern Europe, the magnitude of the differences is very small and the models with glasso covariance

matrices perform slightly better in the majority of grid boxes.

The generally better performance of the models with shrinkage covariance matrices might be a result of the under-dispersive and less-robust behaviour of the models with glasso covariance matrices that was diagnosed in Sect. 4.1.1. This explanation is underlined by the spatial patterns of the performance differences. In western Europe and Fennoscandia, the local reconstructions tend to be less informative than in central and eastern Europe. Therefore, the reconstructions at the respective grid boxes

are less constrained by nearby local reconstruction and more by far away proxy samples. This effect is enhanced in the glasso based models with less spatial degrees of freedom. Therefore, reconstructions with glasso matrices can be more biased if information are inadequately transferred over large differences. In central and eastern Europe this effect is less problematic, as the nearby proxy samples are the main contributors to the reconstructions at the left-out grid boxes. That the glasso based models perform even slightly better in central and eastern Europe could than be an effect of the less dispersive behaviour compared to

the models with shrinkage covariances.

### 4.1.3   Conclusions from the comparison study

The ITEs show that the models with shrinkage matrix covariances are more dispersive, less biased, and more robust than those with glasso covariance matrices. These properties transfer to the CVEs where the models with shrinkage covariance matrix perform better, too. The results from models with the same covariance matrix are very similar except that the KM with

shrinkage covariance matrix is on average more biased than the respective GM and RM. This shows that the covariance matrix choice determines the reconstruction skill more than the general formulation of the process stage as Gaussian, regression, or kernel model.

The better performance of shrinkage covariance models shows that the low number of spatial modes as a result of the small ensemble size is the main reason for the under-dispersiveness of the glasso based models. On the other hand, the over-

dispersiveness of the shrinkage models should be an indicator that this model is not under-dispersed even in real world applications which face additional challenges from potential biases or under-dispersed transfer functions and from a potentially more sophisticated spatial structure of the climate state than in the ESM climatologies. Additionally, this over-dispersiveness shows that in most regions the ensemble spread is wide enough to lead to reconstructions which do not feature too narrow posterior distributions as long as enough spatial degrees of freedom are incorporated in the second moment properties of the process stage.

The larger biases of the KM with shrinkage covariance matrix compared to the GM and RM are a result of ensemble member weight degeneracy in the particle filter part of this model. The ensemble member weights tend to degenerate towards the least wrong model such that the mean values are biased towards that model. This tendency increases with the strength of the proxy data signal. This is a well-known issue of Bayesian model selection (Yang and Zhu, 2018), and therefore as well of particle filter methods (Carrassi et al., 2018), which hinders the use of KMs in data assimilation problems. To mitigate this issue, the particle filter part of the KM is combined with a Gaussian part that is more similar to Kalman type filters. The ITEs show that this adjustment is strong enough to avoid under-dispersiveness, but the degeneracy of the ensemble member weights still leads to larger biases than in the process stage models that rely on direct averaging of the ensemble members.

## 4.2 Spatial reconstruction of European MH climate

Based on the results presented in the previous section, the models with shrinkage matrix should be preferred over those with glasso covariance models. In addition, the smaller biases and more robust nature of the GM and RM with shrinkage covariance matrix compared to the KM model, makes them superior choices. Because the RM adjusts more flexible to the proxy data than the GM, this model is presumably better suited to deal with additional caveats of real world applications compared to the controlled test environment of ITEs. Therefore, this model is used for the spatial reconstructions, whose results are presented in this section. Reconstruction results are summarized in Table 1. Results from reconstructions with the other five process stage models are presented in the Supplement.

### 4.2.1 Posterior mean and uncertainty structure

The spatially averaged mean temperature of the reconstruction (posterior mean) is 18.27°C (90% CI: (17.79°C, 18.75°C)) for MTWA and 1.81°C (90% CI: (1.22°C, 2.45°C)) for MTCO, which is in both cases warmer than the the CRU reference climatology (+0.51 K for MTWA and +0.69 K for MTCO). Larger anomalies are found for subregions (Fig. 7a,b). For MTWA as well as MTCO, temperatures were cooler than today in many southern European areas, while in northern Europe the temperatures were predominantly higher than today. More specifically, MTWA was warmer over Fennoscandia, the British Islands, and the Norwegian Sea. Most of these anomalies are significant on a 5% level. Here, a positive anomaly is called significant if the probability to exceed the reference climatology is at least 0.95. Significant negative anomalies are defined accordingly. The significance estimates are calculated point-wise. Negative MTWA anomalies are found in large parts of the Mediterranean and eastern Europe, but fewer anomalies are significant on a 5% level than in north-western Europe. The largest positive MTCO anomalies are found in Fennoscandia and off the Norwegian coast. In the other parts of the domain, the majority of MTCO

anomalies are negative, but the spatial pattern is more heterogeneous than for MTWA. A lot fewer MTCO anomalies are significant on a 5% level compared to MTWA anomalies. Using the joint information contained in the proxy synthesis and combining it with the spatial structure of the PMIP3 ensemble leads to more significant signals than in the individual data products.

Most of the taxa, which are used in the reconstruction, are stronger confined for MTWA than for MTCO because the growth
of most European plants is more sensitive to conditions during the growing season. This results in more constrained local MTWA reconstructions (Fig. 4c), which is in concordance with findings from Gebhardt et al. (2008). Hence, the uncertainty in the MTWA reconstruction is smaller than in the MTCO reconstruction with spatially averaged point-wise 90% CI sizes of 4.15 K and 5.84 K, respectively (Fig. 7c,d). The uncertainty is smallest at grid boxes with proxy records, and highest in the north eastern and north western parts of the domain where the PMIP3 ensemble spread is large and the constraint from proxy
data is weak. For MTWA, additional regions with large uncertainties are found at the eastern and southern boundaries of the domain due to weak proxy data constraints. Besides, the reconstruction uncertainty has small spatial variations. The ratio of the CI size of spatially averaged temperatures and the spatially averaged point-wise CIs can be interpreted as a measure for the spatial degrees of freedom in the reconstruction.

The highest reduction of uncertainty due to the inclusion of proxy data is found at grid boxes with proxy data, as quantified
by a spatially averaged reduction of point-wise CI sizes from prior to posterior of 50.1% compared to 26.0% for grid boxes without proxy data (Fig. 7e,f). The uncertainty reduction for MTWA is higher for terrestrial grid boxes than marine ones, but the smaller PMIP3 ensemble spread over the British Islands, the North Sea, and the Bay of Biscay leads to similar posterior CI sizes in these areas. For MTCO, the reduction of uncertainty is generally smaller than for MTWA due to the weaker proxy data constraint.

To study whether the degree of spatial smoothing of the reconstruction is reasonable, a measure inspired by discrete gradients is calculated. For each grid box, the mean absolute difference between the value in the box and its eight nearest neighbours is computed. Then, the spatial averages of this homogeneity measure $H$ in the posterior, the climatologies of the PMIP3 ensemble members, and the reference climatology are compared. A reconstruction with a good degree of smoothing is expected to have similar spatial homogeneity than the PMIP3 ensemble and the reference climatology, as $H$ depends mainly on local features
like orography or land-sea contrasts, and we expect these features to affect the local climate of the MH similarly than today's climate since reconstructions suggest only small changes in topography between the MH and today. For MTWA, the posterior mean value is 1.41 K (90% CI: (1.31 K, 1.53 K)), which is in agreement with 1.39 K for the reference climatology and values between 1.08 K and 1.54 K for the PMIP3 climatologies. The heterogeneity of MTCO is higher than of MTWA, but the mean posterior value of 2.54 K (90% CI: (2.33 K, 2.76 K)) is of comparable magnitude as the reference climatology (2.02 K) and
the PMIP3 climatologies (between 1.89 K and 2.41 K). From these results, it is deduced that the posterior has a reasonable degree of spatial smoothing.

### 4.2.2   Comparison of unconstrained PMIP3 ensemble and posterior distribution

By comparing the posterior with the prior and the local reconstructions, it can be seen that for most areas with nearby proxy records the posterior mean resembles the local reconstructions more than the PMIP3 ensemble mean. This shows that the

uncertainty in the prior distribution is large enough to lead to a reconstruction which is mostly determined by proxy data, where available. The posterior MTWA mean is warmer in northern Europe than the prior mean and cooler in southern and eastern Europe. For MTCO, the posterior mean is much warmer than the prior mean in Fennoscandia and slightly cooler in southern Europe.

The posterior weights $\lambda$ of the PMIP3 ensemble members are a combination of the prior distribution of $\lambda$ and the likelihood of $C_p$ for each combination of ensemble member weights (see Appendix C for details). $\lambda$ provides information about which combination of ensemble members fits best to the proxy data. In our reconstruction, the MPI-ESM-P climatology has the highest posterior weights (mean of 0.485) (see Fig. 8), followed by the EC-Earth-2-2 climatology (posterior mean of 0.154) and the Had-GEM2-CC climatology (posterior mean of 0.104). Note, that the weights of the MPI-ESM-P and the EC-Earth-

2-2 are the only one that are on average higher than the prior mean of 1/7. The large differences of the weights are a result of the large differences between the ensemble member climatologies. Because there is less uncertainty in the local MTWA reconstructions, it is the major variable for determining the posterior weights. Among all included models, the MPI-ESM-P simulation is closest to the dipole structure with MTWA warming in northern and cooling in southern Europe, which explains the high model weight.

**4.2.3    Added value of the reconstruction**

CVEs provide inside into the value that is added to the unconstrained PMIP3 ensemble, represented by the process stage Eq. (7), by constraining it with the Simonis et al. (2012) synthesis. To quantify the added value, the BS from Eq. (20) is calculated for the unconstrained process stage, which is called BS(Prior), and compared to the BS of the posterior, BS(Posterior), calculated from leave-one-out CVEs. Then, the Brier skill score (BSS)

$$\text{BSS} := \frac{\text{BS(Prior)} - \text{BS(Posterior)}}{\text{BS(Prior)}} \qquad (21)$$

is computed, which is a measure of the added value of the spatial reconstruction. For positive BSS values, the posterior distribution is superior to the prior. On the other hand, the posterior distribution is inferior to the prior for negative values. This would indicate inconsistencies in the local proxy reconstructions, a scaling mismatch of simulations and data, or the existence of spurious correlations in the prior covariance matrix.

For most left-out proxy samples, the BSS is positive (68.9% of grid boxes) with a median of 0.28 (Fig. 9). The BSS values are predominantly positive for all regions but the British Islands and the Alps. This indicates a high consistency of the reconstruction in large parts of the domain. In particular, consistent MTWA cooling in southern and eastern Europe in the local reconstructions compared to the prior distribution leads to cooling and reduction of uncertainty in the posterior compared to the unconstrained PMIP3 ensemble. Similarly, the consistent MTWA and MTCO warming of the local reconstructions in

the north-eastern part of the domain lead to positive BSS values.

The persistent negative BSS values for the British Islands cannot be explained by data outliers but warrant a systematic issue. For this region, the uncertainty in the local reconstructions is larger than for other areas, such that the local proxy records constrain the posterior less than the posterior ensemble member weights and some of the more distant proxy records. This

leads to a reduction of the posterior uncertainty compared to the unconstrained PMIP3 ensemble but without improving the concordance of the mean state with the local reconstructions, which in turn results in negative BSS values. In and near the Alps, negative BSS might be a result of not accounting enough for orographic effects in the different sources of information.

### 4.2.4   Joint versus separate MTWA and MTCO reconstructions

To study the effect of reconstructing MTWA and MTCO jointly compared to separately, additional reconstructions with only one climate variable are computed. Note that the interactions of MTWA and MTCO are twofold in the joint reconstruction: (a) the response functions have an interaction term in the logistic regression Eq. (3), and (b) the process stage contains joint ensemble member weights for MTWA and MTCO as well as inter-variable correlations in the empirical correlation matrix.

The separate MTWA reconstruction is on average around 0.5 K warmer than the joint reconstruction, whereas the spatially averaged posterior mean of the separate MTCO reconstruction is 0.83 K cooler (Table 3). Hence, the seasonal difference is smaller in the joint reconstruction, due to smoothing from the PMIP3 ensemble and slightly positive correlations between MTWA and MTCO in most of the joint local reconstructions. The MTWA only reconstruction is warmer in most land areas, with largest differences in southern and eastern Europe, but the differences are almost never significant on a 5% level (Fig. 10a). As this part of the domain is best constrained by proxy data, and because the posterior ensemble member weights are similar to the joint reconstruction (mean $\lambda_k$ of 0.419 for MPI-ESM-P), it is likely that the additional warming is due to the missing interaction in the transfer function. On the other hand, the posterior ensemble member weights change a lot for the separate MTCO reconstruction, with HadGEM2-CC and MRI-CGCM3 being the models with the highest weights (mean $\lambda_k$ of 0.426 and 0.199, respectively). Together with the less constrained transfer functions for MTCO than MTWA and the missing interaction term, this leads to a cooler reconstruction for most areas but some parts of the Mediterranean (Fig. 10b). The cooling is strongest in Scandinavia, the British Islands, the Norwegian Sea, and the Iberian peninsula, but almost never significant on a 5% level. As these are the regions which are least constrained by proxy data, choosing different PMIP3 ensemble members affects the reconstruction more than in central Europe, where MTCO is best constrained by proxy data. The reconstruction uncertainties are of similar magnitude in the joint and the separate reconstructions (Table 3), because the interactions in the transfer functions do not reduce the marginal uncertainties and the shrinkage target $\Phi$ does not contain correlations of MTWA and MTCO.

The BSS pattern in the MTWA only reconstruction is mostly the same than in the joint reconstruction except for slightly positive skill in the British Islands (Table 3, Fig. 10e). This shows that the added value of the joint reconstruction compared to the unconstrained PMIP3 ensemble is mainly determined by the MTWA reconstruction. On the other hand, the added value of the MTCO only reconstruction is much smaller (Table 3, Fig. 10f) due to larger uncertainties in the local MTCO reconstructions.

The results show that the more constrained local MTWA reconstructions lead to a higher influence on the joint reconstruction than the local MTCO reconstructions. Therefore, the MTCO only reconstruction differs more from the MTCO estimate in the joint reconstruction. Reconstructing MTWA and MTCO jointly should in theory lead to a physically more reasonable reconstruction by creating samples drawn from the same combination of ensemble members. In the example of this study, this

effect can be seen from the smaller seasonal differences in the joint than in the separate reconstructions. On the other hand, Rehfeld et al. (2016) show that multi-variable reconstructions from pollen assemblages can be biased when signals from a dominant variable are transferred to a minor variable. While the PITM model might be less sensitive to this issue than the weighted averaging transfer function used in Rehfeld et al. (2016) because it better respects the larger MTCO uncertainties and because it is unclear whether taxa occurrence is similarly susceptible to the issue than pollen ratios, it will be subject to future work to study whether joint or separate reconstructions lead to more reliable results.

## 5 Discussion and possible extensions

### 5.1 Robustness of the reconstruction

Our approach is designed with the goal of being more suitable for sparse data situations than standard geostatistical models. To understand the robustness of the Bayesian framework with respect to the amount of data included in a proxy synthesis, five experiments with only half of the samples are performed, which are either selected to retain the spatial distribution of proxy samples or chosen randomly. In all of the tests, the general spatial structure of the posterior distribution, including the anomaly patterns, is preserved, even though depending on the chosen proxy samples the local anomalies and magnitude of changes varies which should be expected when such a large portion of the already sparse data is left out. Only the Norwegian Sea in the MTCO reconstruction changes substantially in some experiments. Plots from the experiments with reduced proxy samples are provided in the Supplement.

The mean spatial averages differ up to 0.6 K for MTWA as well as MTCO, but none of the changes is significant on a 5% level. In contrast, the uncertainty estimates are consistent across all experiments with reduced proxy samples with averaged point-wise 90% CIs that grow by up to 0.4 K from the reconstruction with the full proxy synthesis. Considering that half the proxy samples are left out, this number is low. In all experiments, the spatial homogeneity $H$ is not significantly different from the values reported in Table 3, which shows that the spatial homogeneity as a local feature is more controlled by the process stage than the proxy data. In all but one experiment, the MPI-ESM-P remains the ensemble member with the largest weight $\lambda_k$, and the three ESMs which are favoured neither in the MTWA nor in the MTCO reconstruction retain very low weights in all experiments. But depending on whether proxy samples in which MTCO is much less constrained than MTWA are removed or not, the weights of the four models with the highest values in the joint and separate reconstructions can vary such that in one case the average weight of the EC-Earth-2-2 is 0.1 higher than the weight of MPI-ESM-P. These changes of the weights explain the MTCO changes in the Norwegian Sea as this is the region which is most influenced by the ensemble member weights. The experiments show that the reconstruction is robust with respect to the number of proxy samples as long as the remaining samples are informative and relatively uniformly distributed across space. In our example, this is not the case for the Norwegian Sea. Combining pollen records with sea surface temperature proxies could potentially overcome this issue.

The large PMIP3 ensemble spread for most grid boxes shows that the prior distribution, which is calculated from the ensemble, contains a wide range of possible states. In areas which are well constrained by proxy data, this large total uncertainty leads to a reconstruction which depends little on the climatologies of the ensemble members. Hence, in these areas, the recon-

struction is not sensitive to the particular formulation of the process stage (compare with Supplement). This shows that our method is applicable despite well-known model-data mismatches for the MH (Mauri et al., 2014). On the other hand, the spatial correlation structure controls the spread of local information into space. Different formulations of the spatial correlation matrix can lead to substantially different reconstructions in regions that are not well constrained by proxy samples and in particular a

spatial covariance with too few spatial modes can lead to overly optimistic uncertainty estimates.

## 5.2     Comparison with previous reconstructions

Several reconstructions of European climate during the MH have been compiled previously. Here, we compare our reconstructions to those of Mauri et al. (2015), Simonis et al. (2012), and Bartlein et al. (2011).

Mauri et al. (2015) use a plant functional type modern analogue transfer function and a thin spline interpolation for pollen

samples stemming mostly from the European pollen database. The simpler interpolation method allows the treatment of the samples as point data and the interpolation of the local reconstructions to an arbitrary grid. Among other variables, summer and winter temperatures are reconstructed. We find a dipole anomaly structure similar to Mauri et al. (2015) in our reconstructions, with mostly positive anomalies in northern Europe and negative anomalies in southern Europe. In Mauri et al. (2015) as well as in our reconstruction, the Alps are the only region with significant warming in central and southern Europe for summer

temperature. Generally, the amplitude of summer anomalies in the two reconstructions is similar, although locally there are differences with cooler anomalies over south-western Fennoscandia in our reconstruction as well as warmer anomalies in Finland. For winter temperatures, the cooling in the Mediterranean and the British Islands is less pronounced and spatially less consistent in our reconstruction than in Mauri et al. (2015). As for summer temperatures, we find smaller anomalies in southern Fennoscandia. In contrast, our reconstruction shows higher anomalies in northern Scandinavia.

The same pollen dataset and another version of PITM are used in Simonis et al. (2012) to reconstruct July and January temperature, such that differences between the two reconstructions are mostly related to the different smoothing technique. Simonis et al. (2012) minimize a cost function which combines pollen samples with an advection-diffusion model that is driven by insolation changes between the MH and today. In Simonis et al. (2012), the dipole structure is not found in the same way than in our reconstruction, which might be due to the different way how regions which are not well constrained by proxy

data are treated. Both reconstructions share positive summer temperature anomalies in northern Europe as well as negative anomalies in central Europe and the Iberian peninsula. Unlike our reconstruction, Simonis et al. (2012) find positive anomalies in south-eastern Europe. For winter temperatures, the reconstruction of Simonis et al. (2012) shows an east-west dipole in contrast to the north-east to south-west dipole in our reconstruction. This different structure might be due to the smaller proxy data control of the winter reconstructions, which leads to a higher importance of the interpolation schemes.

A reconstruction designed to evaluate the PMIP3 simulations was provided by Bartlein et al. (2011). They combine a large number of pollen based local reconstructions from the literature to produce a gridded product of six climate variables including MTWA and MTCO. In contrast to our reconstruction, the used local reconstructions are not smoothed across space but only within a grid box. Their results show a dipole structure but less pronounced than in our reconstruction and in Mauri et al. (2015). In particular, they find a cooling for eastern Fennoscandia in summer, a much smaller warming of northern Fennoscandia than

our reconstruction, and a warming in Germany and France. On the other hand, the reported anomalies in Bartlein et al. (2011) for the Mediterranean and eastern Europe are similar to our results.

The comparisons show that patterns like the dipole type anomaly structure, which are not present in the PMIP3 ensemble, seem to be consistent across pollen transfer functions. While some of the differences between the existing literature and our results can be explained by the used transfer functions and proxy syntheses, the choice of an appropriate interpolation method plays an important role, too, especially in areas with very sparse and weakly informative proxy data and to determine the degree of smoothing.

## 5.3 Climate model inadequacy and process stage structure

To account for inadequacies of climate models to simulate past climate states, we introduced flexible ensemble member weights $\lambda$ and the shrinkage matrix approach which combines the empirical covariance matrix of the climate ensemble with a matrix that is derived from an independent physically motivated model. Combining ensemble filtering methods with additional techniques to correct model biases in a physically consistent way is an important but also challenging direction of future work on climate field reconstructions as a balance has to be found between under-dispersion of the posterior distribution by inducing physical structure and overfitting to noisy proxy data by enhancing the degrees of freedom. Beyond the strategies implemented in this work, some directions that can be envisaged are the increase of permitted spatial structures in the prior mean by adding patterns calculated from alternative physically motivated models, and the introduction of multiple shrinkage targets in the spatial covariance matrix (Gray et al., 2018).

In addition to those extensions of the current filtering framework, different types of process stages can be envisaged which are independent of simulations with ESMs but still computed from physical principles. One example are stochastic energy balance models, which are simple stochastic climate models that simulate the energy fluxes between external forcing, atmosphere, and surface. These types of models have become important tools in studies of climate variability Rypdal et al. (2015) and could be reformulated as process stage in BHMs. The advantage of such a process stage is that it can be fully integrated in the BHM instead of using an offline run with an ESM which does not learn from the proxy data. In addition, spatial reconstructions that are independent of ESM output, are more suitable for comparisons of proxy data and climate simulations than the Bayesian filtering approach.

Another valuable extension of our approach would be the computation of reconstructions on hemispheric to global scales. In this case, a more flexible structure for the ensemble member weights is desirable to account for the varying skill of climate models in different regions. On the other hand, the weights should not change too much within small domains to avoid unreasonable spatial heterogeneity. An additional problem is that the stationary shrinkage target matrix would no longer be a good approximation as different regions require different target matrices. This means that non-stationarities have to be integrated in the shrinkage target. A straightforward to accomplish this way is to introduce non-constant parameters in the stochastic partial differential equation behind the Matérn model (Lindgren et al., 2011).

The implementation of these different types of process stages would facilitate more quantitative comparisons of reconstructions and allow a fair testing of modelling assumptions by using ITEs and CVEs. In particular, the reformulation of

existing climate field reconstruction techniques as BHM (Tingley et al., 2012) offers many model comparison techniques that are currently not available as the borders between very different statistical techniques especially to estimate reconstruction uncertainties have to be crossed.

## 6 Conclusions

We presented a new method for probabilistic spatial reconstructions of paleoclimate. The approach combines the strengths of pollen records, which provide information about the climate state in a small scale domain, and of climate simulations, which downscale forcing conditions to physically consistent regional climate patterns. Thus, we reconstruct physically reasonable spatial fields, which are consistent with a given proxy synthesis. Our framework can deal with probabilistic transfer functions, which are non-linear and non-Gaussian, such that an extension to a wide range of proxies and associated transfer functions is

possible.

Using ITEs and CVEs, we showed that robust spatial reconstructions with Bayesian filtering methods that exhibit small biases and are not under-dispersed are possible as long as the statistical framework is flexible enough to account for deficiencies of climate simulations and to avoid filter degeneracy, which can emerge due to small ensemble sizes and biases in climate simulations. However, all these properties can be lost when the model does not adequately account for those two issues. The

resulting model, which is used for spatial reconstructions of European MH climate, uses a weighted average of the involved ensemble member climatologies and a shrinkage matrix approach for spatial interpolation and structural extrapolation of the proxy data. A strong over-dispersion is found in ITEs which is less harmful than under-dispersion because it avoids the drawing of overconfident conclusions.

We apply our framework to reconstruct MTWA and MTCO in Europe during the MH using the proxy synthesis of Simonis

et al. (2012) and the PMIP3 MH ensemble. Brier scores from cross-validations reveal that the spatial reconstruction predominantly adds value to the unconstrained PMIP3 ensemble, and analyses of the spatial homogeneity of the posterior distribution indicate a reasonable degree of smoothing through the induced correlation structure. The large scale spatial patterns of our reconstruction are in agreement with previous work (Mauri et al., 2015; Bartlein et al., 2011). As the posterior mean is more similar to the local proxy reconstructions than to the prior mean for most terrestrial areas, we see that the main role of the sim-

ulation ensemble is to provide a spatial covariance structure and that a reconstruction, which is in line with reconstructions that do not include simulation output, is possible despite well-known model-data mismatches (Mauri et al., 2014). Our framework provides a way to quantitatively test hypotheses in paleoclimatology and to assess the consistency of a given proxy synthesis. This includes the fit with large scale structures, the spatial homogeneity, and the quantitative quality control of the proxy data by identification of potential outliers.

*Code and data availability.* R code for computing reconstructions with the presented Bayesian framework is provided in a Bitbucket repository available under https://bitbucket.org/nils_weitzel/spatial_reconstr_repo. The pollen and macrofossil synthesis is published in Simonis

(2009). It is available as an R list object in the Bitbucket repository. The PMIP3 MH simulations are available in the CMIP5 archives. In this study, they were downloaded from the DKRZ long term archive CERA (https://cera-www.dkrz.de). The modern climate data was downloaded from the University of East Anglia Climate Research Unit, available at http://www.cru.uea.ac.uk/data/. The vegetation data for transfer function calibration was provided by Thomas Litt and Norbert Kühl.

## Appendix A:  Determination of glasso penalty parameter

To determine the glasso penalty parameter $\rho$, we first recognize that for values smaller than $\rho = 0.3$ the resulting matrices become numerically unstable in our application due to the small ensemble size. Five values for $\rho$ were tested: 0.3, 0.5, 0.7, 1.0, and 2.0. Larger values lead to sparser precision matrices and therefore to smaller spatial correlations. This in turn means that the local information from the proxy data is spread less into space. For each of the five parameters, we perform CVEs

with the RM and compare the resulting BS (see Sect. 4.1.2). While the smallest penalty parameters have the best mean BS, the differences are generally small (see Supplement). On the other hand, the influence of the penalty term in Eq. (14) on the overall regression increases from 79.5% for $\rho = 0.3$ to 98.5% for $\rho = 2.0$. Based on these diagnostics, we choose $\rho = 0.3$ for the reconstructions in Sect. 4 to get a numerically stable covariance which performs at least as good as other choices of $\rho$ in CVEs, and is comparably little influenced by the penalty term. The sensitivity of the reconstruction with respect to $\rho$ is further

studied in the Supplement.

## Appendix B:  Shrinkage target matrix

The shrinkage target $\Phi$ is defined on a regular lat-lon grid following the SPDE approach of Lindgren et al. (2011). This approach allows a computationally efficient approximation of Matérn covariances with parameters that are physically motivated in the context of stochastic Laplace equations, which model diffusive transport of a stochastic forcing. Setting $\zeta^2 := \frac{8}{\rho^2}$, the range

parameter $\rho$ is rewritten as a scale parameter $\zeta^2$. Moreover, we let the anisotropy parameter $\nu$ parameterize a diagonal diffusion matrix $\upsilon := \mathrm{Diag}\left(\sin(\nu\pi/2), \cos(\nu\pi/2)\right)$. Then, the stochastic partial differential equation from which $\Phi$ is deduced is given by

$$\left(\zeta^2 - \nabla \cdot (\upsilon\nabla)\right) X(x) = \mathcal{W}(x). \tag{B1}$$

$\mathcal{W}$ denotes white noise, and $X$ is the stationary Gaussian random field that solves Eq. (B1). Discretizing this equation with

linear finite elements, and using a diagonal approximation of the involved mass matrix leads to a GMRF with correlation matrix $\hat{\Phi}$ (Lindgren et al., 2011). We use degree as distance unit on the regular lat-lon grid instead of m which means that the decorrelation length depends on the latitude. While this is counterintuitive on first glance, it better reflects the mostly shorter decorrelation lengths in higher latitudes and leads to a better model fit. $\Phi$ is constructed from $\hat{\Phi}$ by combining spatial correlation matrices of type $\hat{\Phi}$ for each climate variable, that is jointly reconstructed, but with different parameters $\rho$ and $\nu$ for

each climate variable, in a block diagonal structure.

## Appendix C: Full conditional distributions

The Metropolis-within-Gibbs approach samples (asymptotically) from the full conditional distributions of each variable, i.e. the distribution of the variable given all other variables. Some variables are treated block-wise to account for correlations between them, while others are updated sequentially if they are independent from each other or the joint distribution is too complicated for efficient sampling. In this appendix, we detail the conditional distributions that are used for sampling.

To sample the transfer function parameters, we introduce PG distributed augmented variables $\gamma_l^T$, where $T \in T(P)$ and $l = 1, ..., L(T)$ are the number of observations for taxa $T$ (Polson et al., 2013). $\gamma_l^T$ is PG distributed given $\beta^T = (\beta_1^T, ..., \beta_6^T)$ and climate data $C(l) = (C_1(l), C_2(l))$, where $C_1$ and $C_2$ denote MTWA and MTCO. More precisely, the full conditional distribution is given by

$$\gamma_l^T \mid \beta^T, C(l) \quad \sim \quad \mathrm{PG}(n = 1, X_{C(l)} \cdot \beta^T), \tag{C1}$$

$$\text{where } X_{C(l)} := \left(1, C_1(l), C_2(l), C_1(l)C_2(l), C_1(l)^2, C_2(l)^2\right). \tag{C2}$$

Including the Gaussian prior defined in Sect. 3.2, the full conditional of $\beta^T$ is Gaussian distributed:

$$\beta^T \mid P_m^T, P_p^T, \gamma_1^T, ..., \gamma_{L(T)}^T \quad \sim \quad \mathcal{N}\left(V_\gamma X^t \kappa^T, V_\gamma\right), \tag{C3}$$

$$\text{where } V_\gamma := \left(X^t \Gamma X + B^{-1}\right)^{-1}. \tag{C4}$$

Here, $X$ is a matrix with rows $X_{C(l)}$ for $l = 1, ..., L(T)$, $\Gamma$ is a diagonal matrix with entries $\gamma_l^T$, $B$ is the $6 \times 6$ prior covariance matrix of $\beta^T$, and $\kappa^T$ is a vector with entries $\left(P^T(l) - \frac{1}{2}\right)$, where $P^T(l)$ is the presence or absence of taxa $T$ in observation $l$. In our case, $B$ is a diagonal matrix with all values equal to 10. Details on the definition of PG variables and the augmented Gibbs sampler can be found in Polson et al. (2013).

Sampling from $\vartheta$ depends on the specific version of the process stage which is used. In the RM, $\lambda = (\lambda_1, ..., \lambda_K)$ is influenced by its prior and by the Gaussian distribution of $C_p$ given $\lambda$:

$$\lambda \mid C_p \quad \sim \quad \mathrm{Dirichlet}\left(\tfrac{1}{2}, ..., \tfrac{1}{2}\right) \mathcal{N}\left(C_p \,\middle|\, \sum_{k=1} \lambda_k \mu_k, \Sigma_{\mathrm{prior}}\right). \tag{C5}$$

This full conditional does not follow a probability distribution from which we can sample directly. Therefore, a random walk type Metropolis-Hastings update is used for updating $\lambda$.

In the KM, the full conditional of $\omega = (\omega_1, ..., \omega_K)$ is Dirichlet distributed given $z = (z_1, ...z_K)$ and its Dirichlet prior:

$$\omega \mid z \quad \sim \quad \mathrm{Dirichlet}\left(\tfrac{1}{2} + z_1, ..., \tfrac{1}{2} + z_K\right). \tag{C6}$$

Given $\omega$ and $C_p$, $z$ is categorically distributed:

$$z \mid \omega, C_p \quad \sim \quad \mathrm{Cat}\left(\alpha_1, ..., \alpha_K\right), \tag{C7}$$

$$\text{where } \alpha_k := \frac{\omega_k \cdot \exp\left(-\frac{1}{2}\left(C_p - \mu_k\right)^t \Sigma_{\mathrm{prior}}^{-1}\left(C_p - \mu_k\right)\right)}{\sum_{i=1}^K \left(\omega_i \cdot \exp\left(-\frac{1}{2}\left(C_p - \mu_i\right)^t \Sigma_{\mathrm{prior}}^{-1}\left(C_p - \mu_i\right)\right)\right)} \tag{C8}$$

If shrinkage covariance matrices are used, the parameters $(\alpha, \rho, \nu)$ have to be chosen in each MCMC step from one of the $K = 7$ predefined parameter sets. We use a uniform prior, i.e. all seven parameter sets have the same prior probability. Then, the full conditional of $\tau$ which indexes the parameter set, is given by

$$\tau \mid C_p, \hat{\mu} \quad \sim \quad \mathcal{N}\left( C_p \left| \sum_{k=1} \hat{\mu}, \Sigma_{\text{prior}}(\alpha_\tau, \rho_\tau, \nu_\tau) \right. \right), \tag{C9}$$

where $\hat{\mu}$ is given according to conditioning on the process stage parameters of the GM, RM, or KM. We update $\tau$ using an Metropolis-Hastings step with independent proposals, which choose $\tau = k$ with probability $\frac{1}{K}$.

We update $C_p(x)$ for $x \in x_P$ sequentially using random walk Metropolis-Hastings sampling. The set of all grid boxes besides $x$ is denoted by $Y_x$, and let $\Sigma_{\text{prior}}^{-1}(a, b)$ be the block of the inverse covariance matrix containing the rows $a$ and columns $b$. Then, the full conditional distribution of $C_p(x)$ depends on the pollen samples $P_p(s)$ with location $x_s = x$, the climate $C_p(Y_x)$ at the other locations, and process stage parameters $\vartheta$. It does not follow a standard distribution:

$$C_p(x) \mid P_p, C_p(Y_x), \vartheta \quad \sim \quad \mathcal{N}\left( \tilde{\mu}_k(x), \left( \Sigma_{\text{prior}}^{-1}(x, x) \right)^{-1} \right) \prod_{\substack{s \\ \text{with} \\ x_s = x}} \prod_{T \in T(s)} \text{logit}\left( X_{C_p(x)} \cdot \beta^T \right), \tag{C10}$$

where $\tilde{\mu}_k(x) := \hat{\mu}_k(x) - \left( \Sigma_{\text{prior}}^{-1}(x, x) \right)^{-1} \Sigma_{\text{prior}}^{-1}(x, Y_x) \left( C_p(Y_x) - \hat{\mu}_k(Y_x) \right). \tag{C11}$

The step size of the random walk proposals is controlled by the conditional covariance $\left( \Sigma_{\text{prior}}^{-1}(x, x) \right)^{-1}$.

Conditioned on $C_p(x_P)$ and $\vartheta$, $C_p(x_Q)$ follows a Gaussian distribution:

$$C_p(x_Q) \mid C_p(x_P), \vartheta \quad \sim \quad \mathcal{N}\left( \tilde{\mu}_k(x_Q), \left( \Sigma_{\text{prior}}^{-1}(x_Q, x_Q) \right)^{-1} \right), \tag{C12}$$

where $\tilde{\mu}_k(x_Q) := \hat{\mu}_k(x_Q) - \left( \Sigma_{\text{prior}}^{-1}(x_Q, x_Q) \right)^{-1} \Sigma_{\text{prior}}^{-1}(x_Q, x_P) \left( C_p(x_P) - \hat{\mu}_k(x_P) \right). \tag{C13}$

## Appendix D: Metropolis coupled Markov chain Monte Carlo algorithm

As described in Sect. 3.4, the multi-modality of the KM in combination with the high dimensionality of the posterior makes the standard MCMC algorithm very inefficient. In our specific formulation the inefficiency is manifested in a very slow mixing of $z$, because conditioned on $C_p$ the likelihood of choosing a new model $z_k$, from one MCMC step to the next one is very small. This problem could be shifted to other variables by integrating out $z$, but than the conditional Gaussian structure of $C_p$ would be lost which would lead to new challenges for generating efficient MCMC strategies. Therefore, we apply a MC[3] or parallel tempering strategy. The original strategy from Geyer (1991) was adapted to parallel computer architectures by Altekar et al. (2004) and applied in a paleoclimate reconstruction problem by Werner and Tingley (2015).

We run $A$ MCMC chains in parallel, and after every $M$ steps, we use an additional Metropolis-Hastings step to swap the states of the Markov chains $a_1$ and $a_2$ with probability $0 < p_{a_1, a_2} < 1$, where $p_{a_1, a_2}$ is calculated from the Metropolis-Hastings odds ratio. The Markov chains are created by exponentiating the process stage and the data stage by constants $\nu_1 = 1 > ... > \nu_A > 0$. The first Markov chain ($\nu_1 = 1$) asymptotically retains the original posterior distribution for all variables,

whereas the subsequent chains sample from a flatter posterior distribution, in which it is easier to jump from one mixture component to another. Following empirical testing, we run the European reconstructions with $A = 8$ parallel chains, levels $\nu_1 = 1, \nu_2 = 1.25^{-1}, ..., \nu_8 = 1.25^{-7}$, and swaps after every $M = 30$ steps. Pseudo-code for the MC$^3$ algorithm is given in the appendix.

5 *Competing interests.* The authors declare that they have no conflict of interest.

*Acknowledgements.* This work was supported by German Federal Ministry of Education and Research (BMBF) as Research for Sustainability initiative (FONA, www.fona.de) through the Palmod project (FKZ: 01LP1509D). N. Weitzel was additionally supported by the National Center for Atmospheric Research (NCAR). NCAR is funded by the National Science Foundation. We acknowledge all groups involved in producing and making available the PMIP3 multi-model ensemble. We acknowledge the World Climate Research Programme's Working
10 Group on Coupled Modelling, which is responsible for CMIP5. For CMIP5 the US Department of Energy's Program for Climate Model Diagnosis and Intercomparison provides coordinating support and led development of software infrastructure in partnership with the Global Organization for Earth System Science Portals. We thank Douglas Nychka for helpful ideas to speed up the MCMC algorithm. We thank two anonymous referees and the editor for their interesting and helpful comments which facilitated a substantial improvement of this manuscript.

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

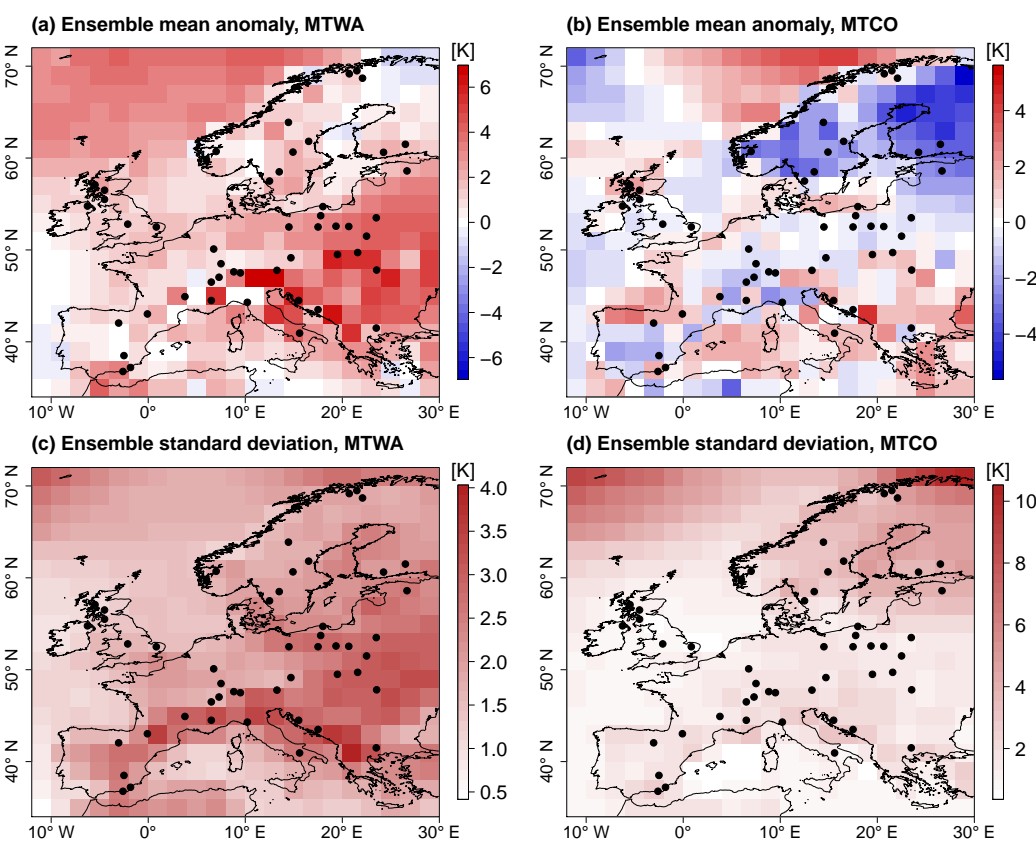

**Figure 1.** PMIP3 MH ensemble mean anomaly from CRU reference climatology, (a) MTWA, (b) MTCO, and ensemble spread as empirical standard deviation of the ensemble members, (c) MTWA, (d) MTCO. Black dots depict proxy samples from Simonis et al. (2012).

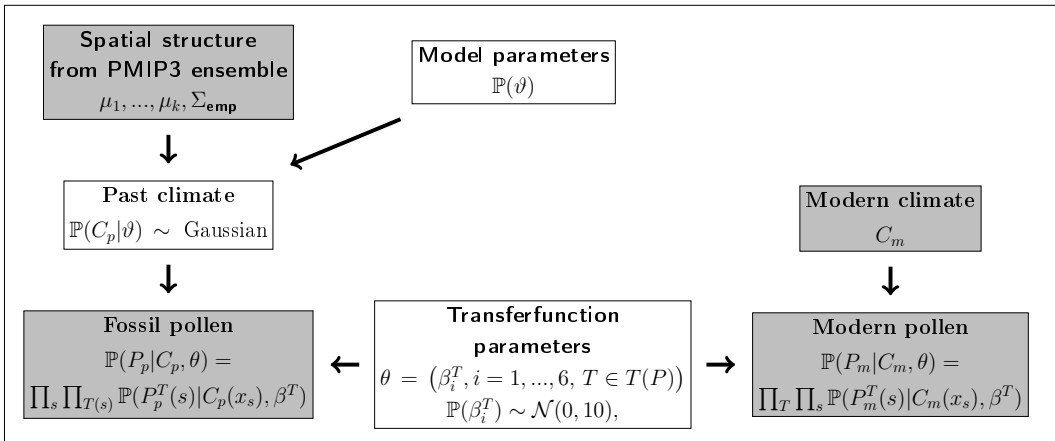

**Figure 2.** Directed acyclic graph corresponding to the Bayesian framework Eq. (2). Involved quantities are given by nodes and arrows indicate dependences of variables. Details of the formulas are explained in Sect. 3.2 and 3.3. White: inferred quantities; gray: input data.

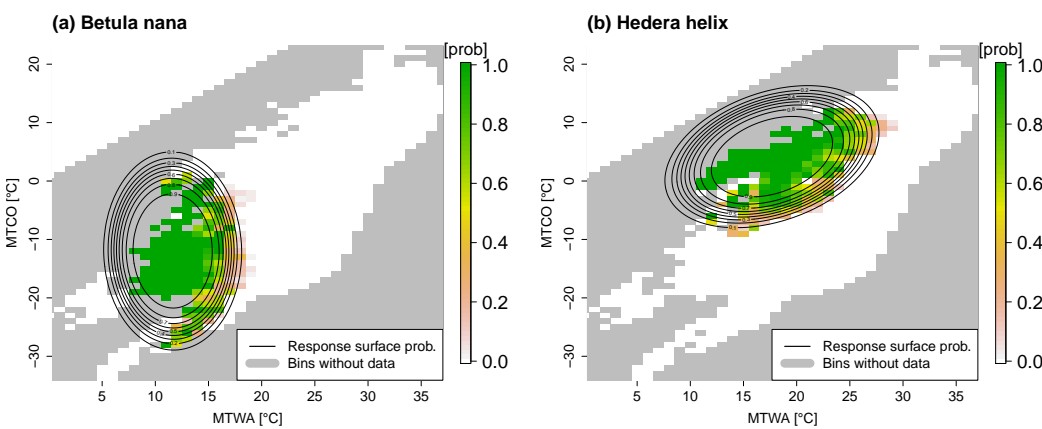

**Figure 3.** Response functions for Betula nana (a) and Hedera helix (b). The relative frequency of occurrence in 1K bins is shown in colours, and the contours depict the probability of presence as estimated by the logistic response function. Gray boxes denote bins without calibration data. In the climate space, combinations of MTWA and MTCO with MTWA < MTCO cannot occur by definition. White bins in the upper left depict artificial absence information added to account for this constraint.

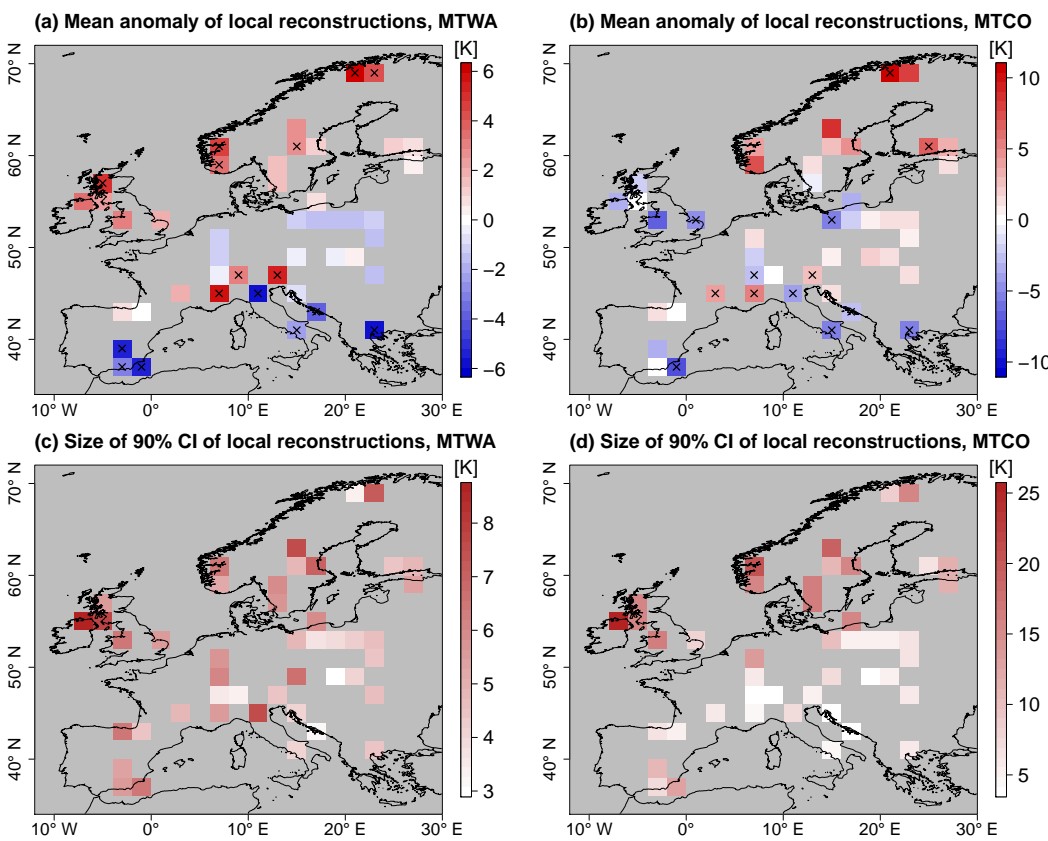

**Figure 4.** Summary statistics of local reconstructions using the PITM forward model. Top row: mean anomaly from CRU reference climatology (left: MTWA, right: MTCO), bottom row: uncertainty measured by the size of marginal 90% CIs (left: MTWA, right: MTCO). In the top row, significant anomalies (5% level) are marked by black crosses.

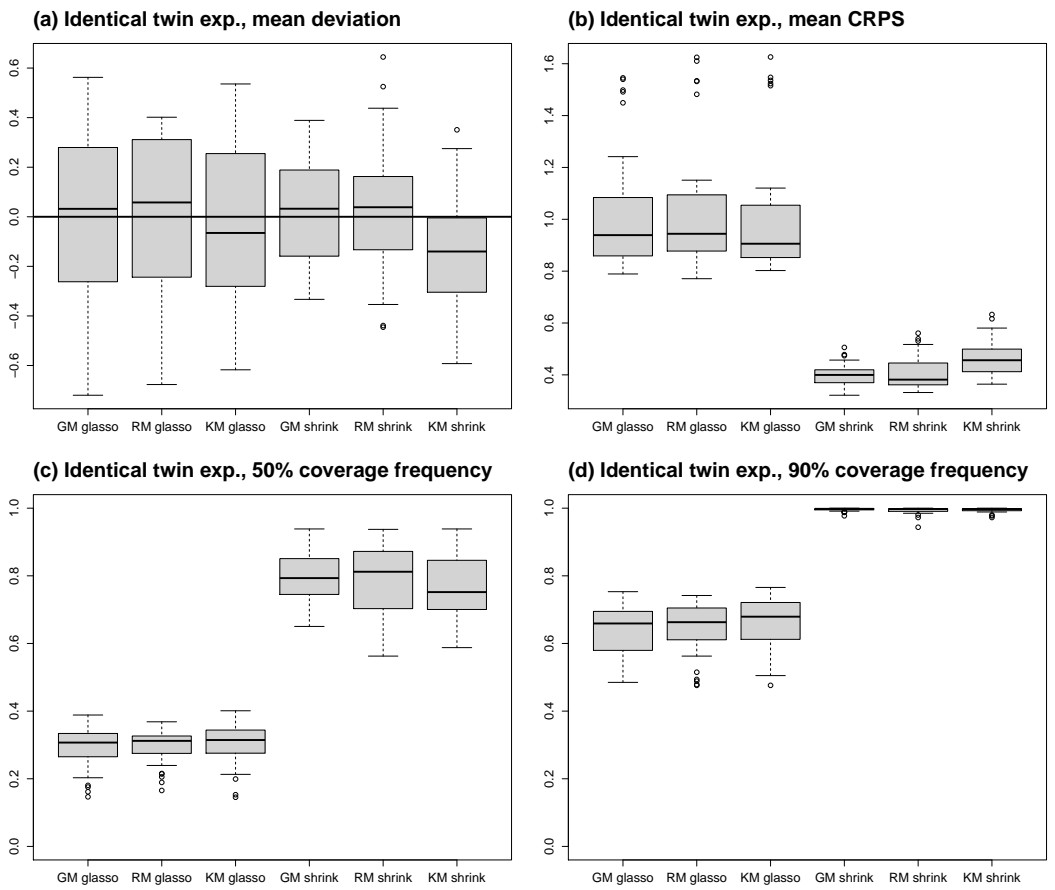

**Figure 5.** Results from ITEs. The boxplots depict the distribution of experiments with the same process stage model. (a) Mean deviation from reference climate, (b) mean CRPS, (c) coverage frequency of 50% CIs, (d) coverage frequency of 90% CIs.

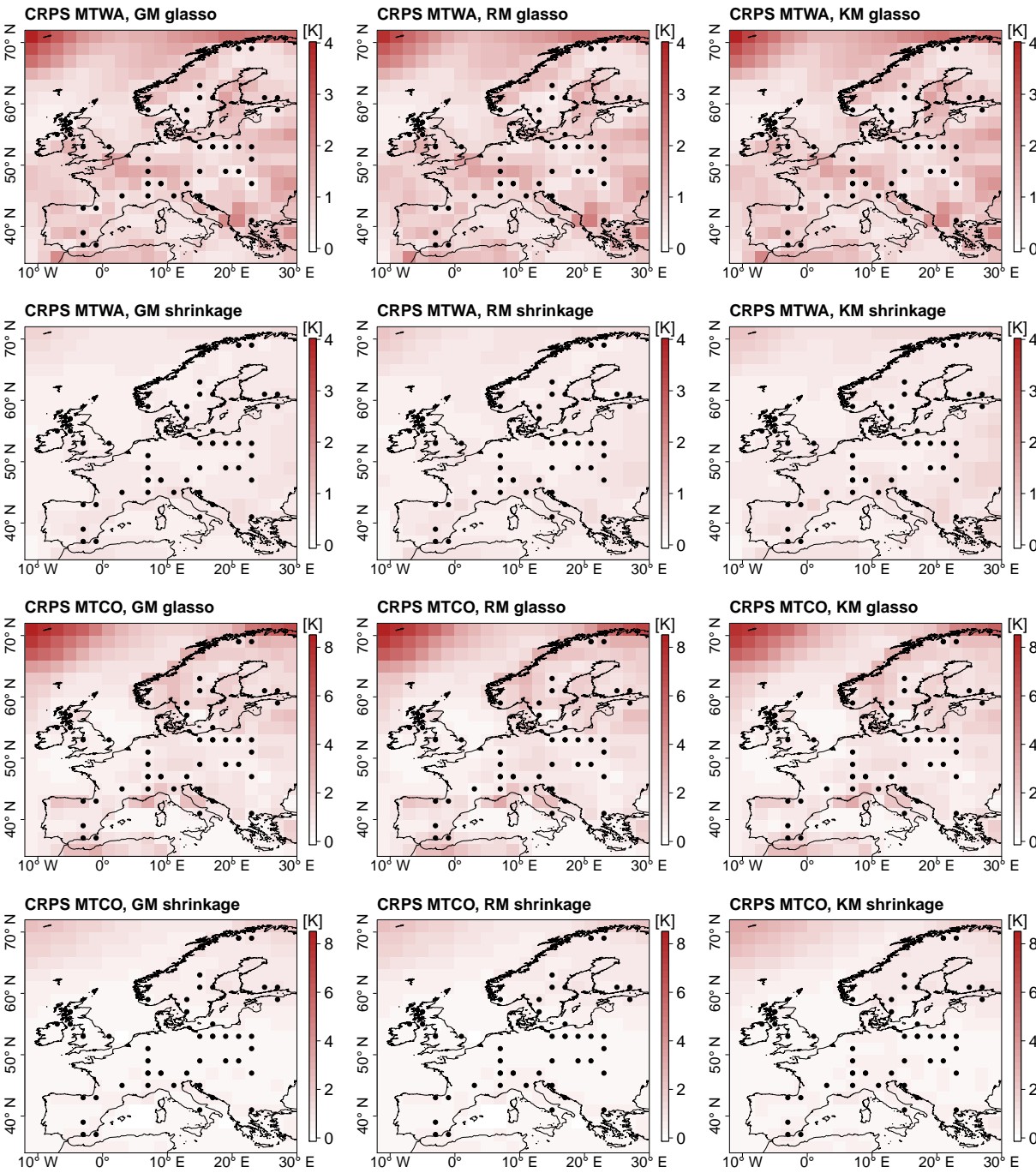

**Figure 6.** Mean CRPS in ITEs for GM, RM, and KM. Top row: Models with glasso covariance matrix, MTWA. Second row: Models with shrinkage covariance matrix, MTWA. Third row: Models with glasso covariance matrix, MTCO, Bottom row: Models with shrinkage covariance matrix, MTCO. Grid boxes with simulated proxy data are depicted by black dots.

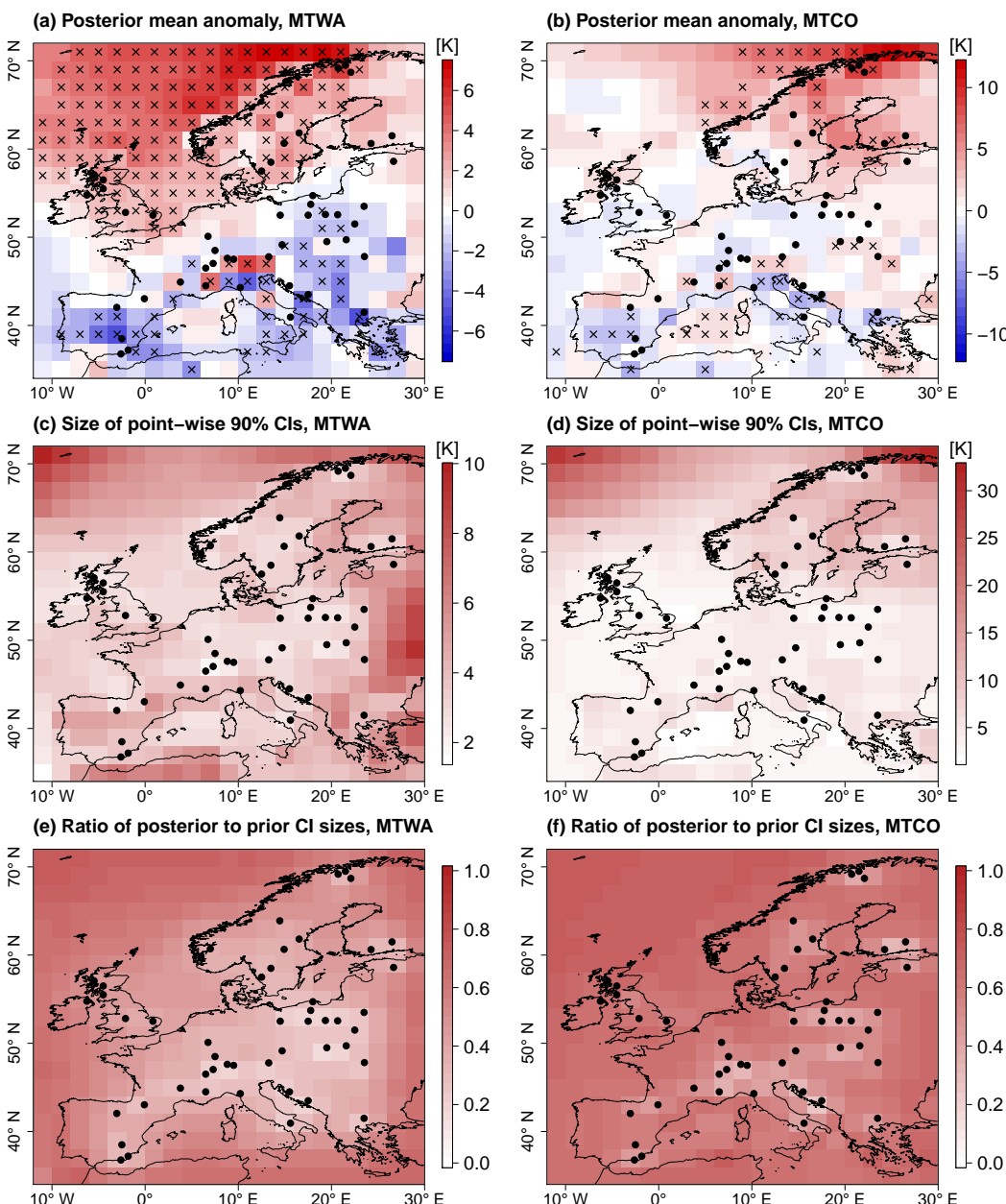

**Figure 7.** Spatial reconstruction for MH. Top row: Posterior mean anomaly from CRU reference climatology (left: MTWA, right: MTCO), middle row: reconstruction uncertainty plotted as size of point-wise 90% CIs (left: MTWA, right: MTCO), bottom row: reduction of uncertainty from posterior to prior measured by ratio of posterior to prior point-wise 90% CI sizes (left: MTWA, right: MTCO). Black dots depict proxy samples. In the top row, point-wise significant anomalies (5% level) are marked by black crosses.

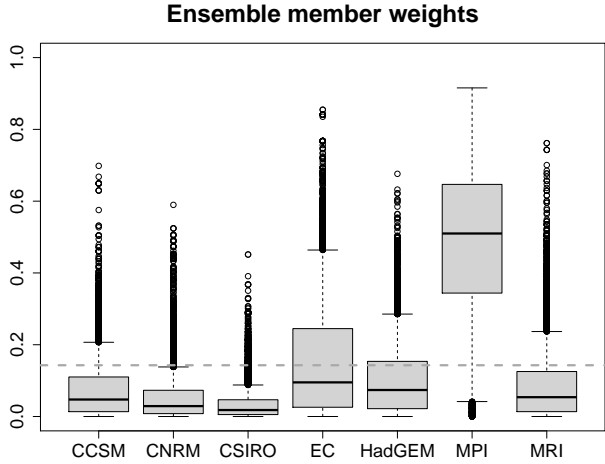

**Figure 8.** Posterior ensemble member weights ($\lambda$) of the reconstruction. Prior weights (mean of $\lambda$) are denoted by the dashed line.

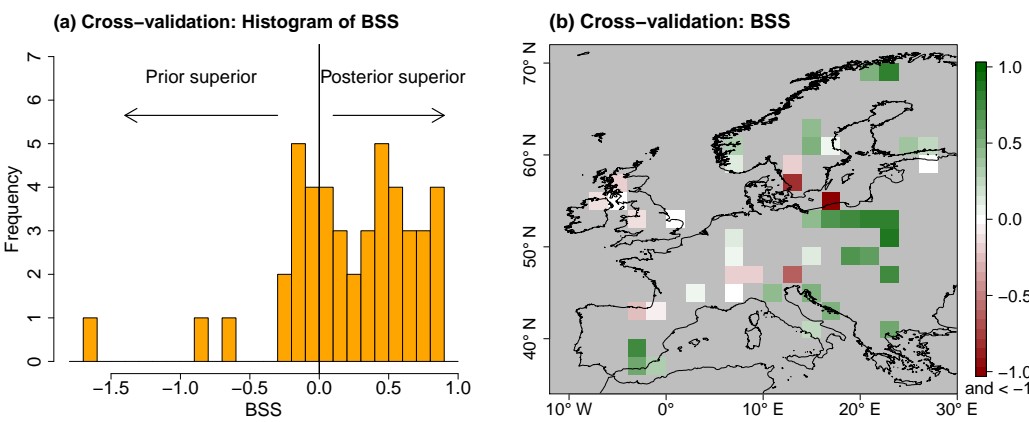

**Figure 9.** BSS from leave-one-out cross-validation. For positive values the posterior is superior to the prior distribution from the unconstrained PMIP3 ensemble, while for negative values the posterior is inferior to the prior. (a) histogram, (b) spatial distribution.

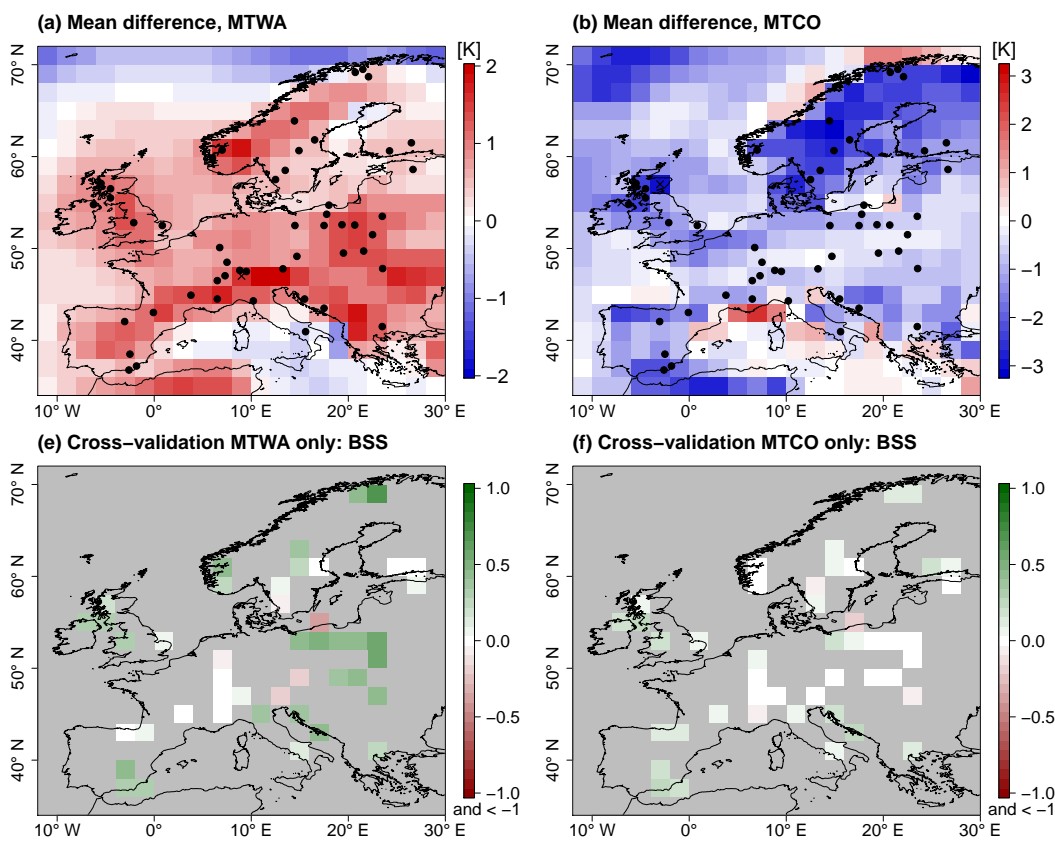

**Figure 10.** Differences of joint and separate reconstructions of MTWA and MTCO. Top row: posterior mean difference (left: MTWA, right: MTCO); bottom row: BSS of the separate reconstructions (left: MTWA, right: MTCO). Black dots depict proxy samples. In the top row, point-wise significant differences (5% level) between the separate and the joint reconstructions are marked by black crosses.

**Table 1.** Basic information on the PMIP3 climate simulations used to construct the process stage in the Bayesian framework (from https://pmip3.lsce.ipsl.fr)

| Model | Institute | Atmospheric grid | Simulated years |
|---|---|---|---|
| CCSM4 | NCAR | 288x192xL26 | 301 |
| CNRM-CM5 | CNRM-CERFACS | 256x128xL31 | 200 |
| CSIRO-Mk3-6-0 | CSIRO-QCCCE | 192x96xL18 | 100 |
| EC-Earth-2-2 | ICHEC | 320x160xL62 | 40 |
| HadGEM2-CC | MOHC | 192x144xL60 | 35 |
| MPI-ESM-P | MPI-M | 196x98xL47 | 100 |
| MRI-CGCM3 | MRI | 320x160xL48 | 100 |

**Table 2.** Summary measures for ITEs and CVEs with the six models described in Sect. 3.3: Spatially averaged mean deviation from reference climatology, spatially averaged size of point-wise 50% CIs, spatially averaged size of point-wise 90% CIs, coverage frequency of 50% CIs, coverage frequency of 90% CIs, spatially averaged CRPS, mean BS.

| | GM glasso | RM glasso | KM glasso | GM shrink | RM shrink | KM shrink |
|---|---|---|---|---|---|---|
| Mean deviation [K] | 0.004 | 0.000 | -0.032 | 0.026 | 0.028 | -0.140 |
| 50% CI size [K] | 1.189 | 1.194 | 1.238 | 1.583 | 1.592 | 1.755 |
| 90% CI size [K] | 2.906 | 2.916 | 3.032 | 3.880 | 3.901 | 4.356 |
| 50% coverage frequency [%] | 29.2 | 29.6 | 30.5 | 79.9 | 78.8 | 76.5 |
| 90% coverage frequency [%] | 64.1 | 64.4 | 66.1 | 99.6 | 99.3 | 99.5 |
| CRPS [K] | 1.03 | 1.032 | 1.010 | 0.399 | 0.408 | 0.468 |
| BS [$p^2$] | 0.186 | 0.186 | 0.187 | 0.165 | 0.163 | 0.161 |

**Table 3.** Summary measures for the joint MTWA and MTCO reconstructions (rows 1 and 2) and the separated reconstructions of MTWA (row 3) and MTCO (row 4). Numbers in brackets are minima and maxima of the corresponding 90% CIs. Additional explanations for all the columns can be found in Sect. 4.2.1 to 4.2.4.

| Reconstruction name | Spatial mean anomaly | Spatially averaged 90% CI size | Point-wise uncertainty reduction | Spatial homogeneity | Median BSS | PMIP3 model with highest weight |
|---|---|---|---|---|---|---|
| Joint reconstruction (MTWA) | (0.03 K) 0.51 K (0.99 K) | 4.15 K | 38.1% | (1.31 K) 1.41 K (1.53 K) | | |
| Joint reconstruction (MTCO) | (0.10 K) 0.69 K (1.32 K) | 5.84 K | 19.6% | (2.33 K) 2.54 K (2.76 K) | 0.28 | MPI-ESM-P |
| Separate MTWA reconstruction | (0.55 K) 1.04 K (1.51 K) | 4.14 K | 38.1% | (1.29 K) 1.41 K (1.51 K) | 0.27 | MPI-ESM-P |
| Separate MTCO reconstruction | (-0.82 K) -0.14 K (0.60 K) | 5.72 K | 20.6% | (2.45 K) 2.69 K (2.92 K) | 0.05 | HadGem2-CC |