# Peer review of "Combining a pollen and macrofossil synthesis with climate simulations for spatial reconstructions of European climate using Bayesian filtering"

_Climate of the Past, 2018_

## Referee Comment (RC1) · Anonymous Referee #1 · 22 Sep 2018

**Review: Combining a pollen synthesis and climate simulations for spatial reconstructions of European climate using Bayesian modelling**

In general, the paper proposes a Bayesian filtering of climate model ensembles using pollen data. The ideas and presentation are interesting and useful, although there are some areas for improvement of both the statistical and computational methods. The filtering of climate to spatial locations with associated uncertainty is very important for secondary analyses (e.g. vegetative growth models, etc.) and provides a useful contribution to the literature. In what follows, I propose some refinements to the statistcial methods presented in the paper.

A few thoughts:

A) What is the meaning of the covariance $\boldsymbol{\Sigma}$? if it really is the inter-model variability, why mix over $\omega_k N(\boldsymbol{\mu}_k, \boldsymbol{\Sigma})$ instead of just using the distribution $N(\bar{\boldsymbol{\mu}}, \boldsymbol{\Sigma})$? What is the meaning of this distribution that your are mixing over? $\boldsymbol{\Sigma}$ is the covariance of the model ensemble, not a particular realization $\boldsymbol{\mu}_k$, thus the distribution you propose doesn't make much sense to me as a meaningful statsitical object. I'm open to being convinced that this idea makes sense, but right now I don't see the purpose.

Also, using this mixing distribution is likely to eliminate the spatial autocorrelation in the $\boldsymbol{\mu}_k$ as neighboring locations are now no longer draws from the same climate model. Instead, it might be simpler and make more sense to mix over $\omega_k \boldsymbol{\mu}_k$. This mixing distribution is over the climate model ensemble and will preserve the spatial autocorrelations in the climate models. Ultimately, I'm not convinced that this covariance is what you want to model.

B) I would also suggest the title should be ". . . using Bayesian Filtering" rather than "Bayesian Modelling." You aren't modeling the climate, just assimilating the proxy data to filter the climate model ensemble. For instance, if the proxies suggest a temperature higher than all of the climate models, then the best your framework can do is the highest temperature from the climate model ensemble (plus a little error from $\boldsymbol{\Sigma}$). Hence, your models are only as reasonable as the climate models (which are likely not great estimates for a given time and location. . . ).

C) The fit of the pollen data by a normal distribution is really poor. The fitted distributions in Figure 3 look nothing like the data distributions. Perhaps a better model is needed. In addition, if you are only using presences in the fossil pollen, your calibration is biased. It would be better to treat absences as a zero-inflated model where an absence could be a true absence or a missing presence due to non-climatic reasons. This is easily done by introducing a latent variable (like you did for the $z$). In the ecological literature on occupancy modeling, this is known as detection modeling (MacKenzie et. a. 2002)

D) Equation 13: If mixing over the z's is the problem, why not integrate/marginalize them out? You can always recover them using composition sampling later. It seems like an awfully complex computational framework (MC^3) for such a simple model framework that could be fixed simply by marginalization.

E) Maybe I missed it but what is the size of the model ensemble $K$ and the number of calibration sites?

F) How to you evaluate the Brier score when you don't include the absences? This seems to introduce a bias and could make the Brier score improper (which limitis its usefulness in comparing models).

**Journal Review points**

1) Scientific significance: Does the manuscript represent a substantial contribution to scientific progress within the scope of Climate of the Past (substantial new concepts, ideas, methods, or data)?

Yes, the manuscript demonstrates the use of climate proxies in model ensemble filtering.

2) Scientific quality: Are the scientific approach and applied methods valid? Are the results discussed in an appropriate and balanced way (consideration of related work, including appropriate references)?

There are some questions about the implementation of the statistical model that are not completely resolved. (particularly the mixing distribution for climate doesn't make sense and the lack of absence data introduces bias in the estimates). The comments above can provide some guidance in resolving these issues.

3) Presentation quality: Are the scientific results and conclusions presented in a clear, concise, and well-structured way (number and quality of figures/tables, appropriate use of English language)?

The paper is reasonably well written from a technical perspective, although more motivation of why particular methods/equations are chosen would be useful. In other words, there is a lot written about *what* the methods are by not much about *why* the methods are chosen and what the ideas are trying to solve.

1) Does the paper address relevant scientific questions within the scope of CP?

Yes

2) Does the paper present novel concepts, ideas, tools, or data?

Yes

3) Are substantial conclusions reached?

Yes. A nice idea for Bayesian filtering is introduces

4) Are the scientific methods and assumptions valid and clearly outlined?

Not always (at least for the statistical methods).

5) Are the results sufficient to support the interpretations and conclusions?

Yes.

6) Is the description of experiments and calculations sufficiently complete and precise to allow their reproduction by fellow scientists (traceability of results)?

For the most part. A gitHub repository/code base would go a long way.

7) Do the authors give proper credit to related work and clearly indicate their own new/original contribution?

Yes.

8) Does the title clearly reflect the contents of the paper?

Yes, with a small change of emphasis

9) Does the abstract provide a concise and complete summary?

Yes.

10) Is the overall presentation well structured and clear?

Yes.

11) Is the language fluent and precise?

Yes.

12) Are mathematical formulae, symbols, abbreviations, and units correctly defined and used?

In general, yes.

13) Should any parts of the paper (text, formulae, figures, tables) be clarified, reduced, combined, or eliminated?

14) Are the number and quality of references appropriate?

Yes

15) Is the amount and quality of supplementary material appropriate?

I would like to see more done for reproducibility. The computational methods seem overly complex and making code availalbe for replication would be useful.

**References**

MacKenzie, D. I., Nichols, J. D., Lachman, G. B., Droege, S., Andrew Royle, J., & Langtimm, C. A. (2002). Estimating site occupancy rates when detection probabilities are less than one. Ecology, 83(8), 2248-2255.

---

## Referee Comment (RC2) · Anonymous Referee #2 · 12 Nov 2018

This is an interesting manuscript which presents a sophisticated Bayesian approach to model-data synthesis. I have some minor criticism of the presentation which often appears poorly-organised to me. Specifically, many parameters and concepts are used prior to their definition and introduction which made the manuscript tough going for me. That is easily fixable of course.

A more fundamental concern I have is demonstrated in the results, which (unless I have misunderstood something) show a strong degree of instability and overconfidence. This is a common occurrence in situations where all members of a model ensemble are inadequate and a Bayesian framework is used which does not allow for

this - for instance, by the explicit use of model inadequacy or model error terms. The z weight of 0.98 when all data are used is itself likely to indicate a problem, and this feeling is only strengthened by the fact that the weight on this model was essentially 0 when half the data were used, in which event two different models are preferred. What seems to be happening here is that the method is homing in on the model(s) for which the data are most likely, while in fact these data are incompatible with any model.

I've seen this in a variety of contexts and am sure this sort of phenomenon will be familiar to the authors. A trivial demonstration of the phenomenon can be given by considering a situation in which we have two models which generate data according to $N(0,1)$ and $N(10,1)$ respectively, whereas the data are in fact generated by $N(5,1)$. A naive filtering which ignores the possibility of model inadequacy will assign essentially all weight to whichever model happens to be closer to the sample mean of the data set (which is of course close to, but not precisely, 5). As more data are sampled, the weight will jump randomly from one model to the other with extremely high confidence, rather than converging to the answer that the models are equiprobable but poor, which might be found if a reasonable model error term was considered. This issue seems to me to undermine the results and details of the application in a fairly fundamental manner but I would hope that the authors could revise their method to adequately account for it, ie by accounting for model inadequacy in a formal manner.

Related to this, the Brier score analysis seems to indicate that the posterior is generally closer to the data than the prior, but it does not indicate whether the posterior is valid in the sense of having reasonably calibrated uncertainties. Also, while the data are in taxa-space, the aim of the paper is actually to recreate a climate, so it is important that the validity of this estimate is assessed. Perhaps a cross-validation (in which one model is used as a target) could be tried.

In summary, I don't want to reject the paper, it is interesting and useful and presents an attractive framework for paleoclimate state estimation. However, I am not convinced that the details of the application as presented here are adequate, nor, therefore, can

I believe that their results are credible. I urge them to modify and extend their work in order to address this.

Minor comments:

As I mentioned, the ordering of some of the paper made it hard work for me.

The prior is introduced in Sect 2.2 and Fig 1 but only explained in Sect 3.2. The variables w and z are first mentioned in 3.1, but for their definitions we are referred forward to 3.2. It is only after substantial usage of the variables, right at the end of 3.2, that we are informed that z was only introduced as a help parameter, with its definition and explanation again pushed forward another two sections to 3.4 with the largely unrelated discussion of the transfer function (two words for this please, and note also that burn-in should be hyphenated) intervening in section 3.3. I would have found it far more straightforward to have had the variables explained as they were introduced.

The multinomial with n=1 would I think be better described as the categorical distribution.

In 3.4, "alternately" usually refers to two alternates, not a sequence. "Sequentially" might be better.

---

## Author Comment (AC1) · 10 Dec 2018

We thank the referee for taking the time to review our discussion paper, and for helpful and interesting comments which will improve the quality of our manuscript. As a response to the referee's suggestions, we plan substantial changes of the manuscript. In the following, referee's comments (RC) are given in blue italicised text and followed by our respective responses (AR).

*"What is the meaning of the covariance $\Sigma$? if it really is the inter-model variability,*

[Figure]

*why mix over $\omega_k N(\mu_k, \Sigma)$ instead of just using the distribution $N(\bar{\mu}, \Sigma)$? What is the meaning of this distribution that your are mixing over? $\Sigma$ is the covariance of the model ensemble, not a particular realization $\mu_k$, thus the distribution you propose doesn't make much sense to me as a meaningful statsitical object. I'm open to being convinced that this idea makes sense, but right now I don't see the purpose.*
*Also, using this mixing distribution is likely to eliminate the spatial autocorrelation in the $\mu_k$ as neighboring locations are now no longer draws from the same climate model. Instead, it might be simpler and make more sense to mix over $\omega_k \mu_k$. This mixing distribution is over the climate model ensemble and will preserve the spatial autocorrelations in the climate models. Ultimately, I'm not convinced that this covariance is what you want to model."*

The general motivation for using the kernel mixture distribution is that each ensemble member is seen as a sample from an unknown distribution of all possible climate states. Because of limitations of the climate models and the small amount of available simulations this is of course a very small subset of possible states. We agree with the referee that ideally one would like the covariance matrix $\Sigma_k$ of each kernel $\mu_k$ to correspond to the climate model $k$ such that the spatial autocorrelation of model $k$ is preserved by $\Sigma_k$ and such that a draw from kernel $k$ is a draw from climate model $k$. Unfortunately, for each model there is only one run available for the mid-Holocene and the internal variability in those runs is much smaller than the inter-model differences. Therefore, using the internal variability of those runs would lead to very distinct kernels and therefore the range of possible states would be very small. Another possibility would be to use long PI control runs, from which more samples could be extracted but still the problem persists that in these runs the internal variability is much smaller than the inter-model differences. Therefore, we estimate $\Sigma$ from the inter-model differences as a compromise that allows to sample from a much broader range of states even though autocorrelation from the individual models is lost. To our knowledge, using the empirical covariance of the samples is a very common choice in kernel based probability density approximations (Silverman, 1986, Chapter 3 and 4) if there is no good estimate of the covariance corresponding to each sample available. Two advantages of mixing over $\omega_k \mathcal{N}(\mu_k, \Sigma)$ compared to using simply $\mathcal{N}(\bar{\mu}, \Sigma)$ are that we do not have to assume that the unknown prior distribution is Gaussian and that we do not rely on an iid assumption for the first moment properties of the kernels. In other words, for the first moment properties of the spatial prior distribution, we do not assume that the model runs are iid samples from a Gaussian distribution but just that there exists an abstract probability density of possible states and that the kernel corresponding to each $\mu_k$ is Gaussian. On the other hand, we do rely on an iid assumption for the second moment properties, which is the compromise we make as described above, due to not seeing a better alternative unless one prescribes second moment properties that are not calculated from climate models. In a meteorology context, the advantages of kernel filters compared to standard Gaussian filtering are described in Anderson and Anderson (1999). A more recent application of kernel filtering in data assimilation is Liu et al. (2016). In both examples, the sample covariance is used as covariance of the samples.

The referee suggests two other models for the spatial prior distribution which would make the inference procedure easier, namely using a Gaussian distribution with the ensemble mean as mean and the inter-model variability to estimate the covariance, i.e. $\mathcal{N}(\bar{\mu}, \Sigma)$, or mixing just over $\omega_k \mu_k$, i.e. $\mathcal{N}(\sum_{k=1}^{K} \omega_k \mu_k, \Sigma)$. Both models have advantages and disadvantages that we want to mention.

Using $\mathcal{N}(\bar{\mu}, \Sigma)$ is the most common approach in data assimilation applications in climatology. It is based on the assumption that the ensemble members are iid samples from an unknown Gaussian distribution which contains all possible climate states. The main advantage of this model is that inference becomes much simpler because the prior distribution is unimodal and Gaussian and not multi-modal as in the kernel approach that we applied. The disadvantage is that it relies on the very strong assumption that the samples $\mu_k$ are iid samples from an unknown Gaussian distribution. This assumption tends to be more realistic for samples from just one climate model, whereas statistics of multi-model ensembles are often not well described by purely Gaussian distributions (Knutti et al., 2010). A second disadvantage of this model is that it reduces the degrees of freedom in the model compared to the kernel model, and therefore limits the possibilities of the model to adjust the posterior distribution to the data. This could be particularly important due to the small ensemble size in our application.

The prior distribution $\mathcal{N}(\sum_{k=1}^{K} \omega_k \mu_k, \Sigma)$ has the advantage that similar to the kernel approach, it is not relying on an iid assumption of the samples for the first moment properties of the distribution. In addition, it introduces more degrees of freedom, as it allows weighted averages of the ensemble members beyond choosing individual members with a certain probability. In addition, the inference becomes easier compared to the kernel approach as this model allows smooth transitions between the $\mu_k$ such that the $MC^3$ (parallel tempering) method is not required to achieve efficient sampling. Similar models are popular in postprocessing of climate model ensembles as in many applications weighted averages outperformed each individual ensemble member (Krishnamurti et al., 1999). A disadvantage of this type of mixing is that even more spatial structure of the $\mu_k$ could be lost compared to the kernel approach because in addition to using the inter-model covariances, the prior mean is a linear combination of the different climate models and does not represent just one single model.

Summarizing, there are good arguments to use each of the three models, the one that we proposed as well as the two suggested be the referee. Similarly, each of the models has disadvantages. Therefore, we decided to test all three models. We additionally compared the performance of using the empirical covariance matrix regularized by the

glasso algorithm with a shrinkage approach, where the empirical correlation matrix is combined with a Matérn type correlation matrix. This is an approach to account for potential model inadequacies to explain the data (see the response to Referee 2). By estimating the shrinkage weight $\alpha$ from the proxy data, this allows deviations from the spatial structures prescribed by the climate simulations if the spatial modes of the Matérn type correlation matrix fit the proxy data better than the inter-model differences.

To compare these six different models, we use cross-validation experiments as described in Sect. 3.5 and 4.2. In addition, we perform identical twin experiments (also named pseudo-proxy experiments or observation system simulation experiments), where one climate model is left out as reference climatology, pseudo-proxies are simulated from this reference climatology and the skill of the corresponding posterior distribution to predict the reference climatology is analysed (see also the response to Referee 2). Initial results suggest that the three statistical models where the glasso regularized covariance matrix $\Sigma$ is used, perform on a similar level, with almost equal Brier scores in the cross-validation experiments and similar skill in the identical twin experiments. On the other hand, the statistical models with the shrinkage covariance matrix outperfom the glasso regularized covariance models in cross-validation as well as identical twin experiments. The main reason for this better performance seems to be a strong reduction of the underdispersion of the posterior distribution due to the increase of spatial modes in the covariance matrix.

In the revised manuscript, we will enhance the motivation for our statistical models. In addition, we will discuss the three different statistical models, the one which we proposed in the manuscript as well as the two that the referee suggested. Moreover, we will describe the two types of covariance matrices, which we compared as a response to the suggestions of the referees. A section on the comparison of the statistical models based on cross-validation and identical twin experiments will be
added. We hope that these changes will not just improve the results of this study but can also provide guidelines for future applications of Bayesian filtering methods in paleoclimate applications.

*"I would also suggest the title should be "... using Bayesian Filtering" rather than "Bayesian Modelling." You aren't modeling the climate, just assimilating the proxy data to filter the climate model ensemble. For instance, if the proxies suggest a temperature higher than all of the climate models, then the best your framework can do is the highest temperature from the climate model ensemble (plus a little error from $\Sigma$). Hence, your models are only as reasonable as the climate models (which are likely not great estimates for a given time and location...)."*

We agree with the referee that the current manuscript title is slightly misleading and that replacing "Bayesian modelling" by "Bayesian filtering" is a more accurate description of our study. Therefore, we will change the title according to the suggestion of the referee in the revised manuscript.

*"The fit of the pollen data by a normal distribution is really poor. The fitted distributions in Figure 3 look nothing like the data distributions. Perhaps a better model is needed."*

While we agree with the referee that there is room for improvement of the response surfaces parametrized by Eq. (7) in the manuscript, we do not think that the fit is "really poor". But maybe the visualisation of the response surfaces in Fig. 3 can be improved. To underline that the chosen parametrization is a reasonable approximation of the probability of presence of the taxa, we plot an alternative to Fig. 3 at the end

of this response. In that plot the coloured raster represents the ratio of taxa presence in bins of size 1K $\times$ 1K, and the contour lines visualize the fitted response surface. Considering that the imperfect sampling of the climate space by the calibration dataset leads to weaker signals in some parts of the climate space, particularly at the edges of the sampled area of the climate space, we think that our current parametrization is a reasonable choice. Therefore, we do not plan to change the parametrization of the response surfaces given by Eq. (7) in the revised manuscript but will look at options to improve the visualisation of the data.

*"In addition, if you are only using presences in the fossil pollen, your calibration is biased. It would be better to treat absences as a zero-inflated model where an absence could be a true absence or a missing presence due to non-climatic reasons. This is easily done by introducing a latent variable (like you did for the z). In the ecological literature on occupancy modeling, this is known as detection modeling (MacKenzie et. a. 2002)"*

The idea of the indicator taxa method is to use taxa which are sensitive to certain climate variables to constrain past climate based on the presence of a taxa in a fossil sample. The probabilistic indicator taxa method (PITM) is an extension of this method where probability distributions have been used to characterize the climate space at which a taxa occurs instead of using binary limits (e.g. a taxa occurs above a certain temperature but not below it) to acknowledge that most taxa have a preferred climate space but the transitions between climates where they usually occur and those where they do not grow is soft. This extension was named pdf method in the literature (Kühl et al., 2002). To estimate these distributions, vegetation data is used instead of modern pollen data, because it contains more accurate information on the presence or absence of a taxa on the spatial scales that we are interested in. As
our Bayesian model is more straightforward formulated by modelling vegetation given climate (forward model) instead of climate given vegetation (inverse model), we have rewritten the pdf method as a forward model by fitting the quadratic logistic regression model, but the aim of this reformulation is to imitate this well-tested method as closely as possible because an extension or improvement of this method is beyond the scope of this study. Because of the reliability of the modern vegetation and climate data, we use presence and absence data to fit the logistic regression. The disadvantage of using vegetation data for the calibration is that the probability of presence of a taxa is only valid in vegetation space on the spatial scale taken for the training data but not in the pollen or macrofossil space, where an absence of a taxa in a pollen or macrofossil sample can have multiple non-climatic reasons like local plant competition or pollen transport effects, as well as local climate effects below the resolution of our study such that the taxa did not grow in the immediate surrounding of the sample. Therefore, the only reliable information on the presence or absence of a taxa in the respective spatial domain (grid box) in the past is the occurrence of the taxa in a pollen or macrofossil sample. Hence, we only use presence information in the reconstruction step.

We agree with the referee that using only present taxa in the fossil pollen is inconsistent with the modern calibration. Despite this inconsistency, our reconstructions are in agreement with previous versions of the probabilistic indicator taxa method, where this inconsistency with the calibration did not appear as previously only presence information were used to fit the probability density functions.

However, we do not see a simple solution for the problem that our calibration is in vegetation space whereas the absence of taxa is an information in the pollen or macrofossil space. The referee suggests to model the absence due to non-climatic reasons as a zero-inflated model by adding a latent variable to estimate the detection probability of a taxa. We think that this is a very promising idea but while the

formulation of this model is simple, the estimation of the detection probability is a very challenging task because it depends on many factors like pollen influx area of the fossil sample, local topography, soil properties, and plant competition which might change over time. It is a priori unclear which of these factor can be marginalized and whether a single detection probability for each taxa is a reasonable approximation. In addition, our fossil dataset combines macrofossils with pollen. The processes that influence the detection probability of macrofossils are very different than for pollen. Therefore, a different detection probability has to be estimated for pollen than for macrofossils.

To our knowledge, the detection probability has never been estimated explicitly as a probability of a presence of a taxa in a pollen or macrofossil sample given the occurrence of the taxa in the respective grid box, but only as a combination of climate as well as non-climate related zero-inflation (e.g Salter-Townshend and Haslett, 2012). We acknowledge that extending the indicator taxa method to include presence and absence information in fossil pollen and macrofossil samples by modelling detection probabilities of taxa should be a focus of future research. But resolving all the described issues requires extensive cooperation of (paleo)climatologists, (paleo)botanists, and statisticians, and is beyond the scope of this study.

To acknowledge the comment of the referee, we will change the following in the revised manuscript: We will describe the underlying assumptions of the PITM model more detailed and elaborate on the inconsistency between the calibration and reconstruction procedure. In addition, we will mention the modelling of detection probabilities as a topic for future research and name the involved issues that need to be solved to accurately model detection probabilities. Finally, we will point out more explicitly that the proxy dataset contains pollen and macrofossil data and the differences of those two data types.

*"Equation 13: If mixing over the z's is the problem, why not integrate/marginalize them out? You can always recover them using composition sampling later. It seems like an awfully complex computational framework (MCЁĘ3) for such a simple model framework that could be fixed simply by marginalization."*

The reason for using the $MC^3$ (parallel tempering) framework is not the mixing of $z$ but the multi-modality of the prior distribution Eq. (6). The poor mixing of $z$ in our case if we do not use $MC^3$ is just the manifestation of that problem. It is widely acknowledged in the literature that the design of efficient MCMC methods for multi-modal models is a challenging task, in particular in multivariate settings. The main issue is the construction of efficient proposal samples, which explore the distribution of the individual modes and jump from one mode to another. $MC^3$ (parallel tempering) is a common technique to solve this issue (Tawn and Roberts, 2018).

Marginalization of $z$ just shifts the general issue to another part of the inference algorithm, as it makes the creation of efficient proposal samples for $C$ a lot more complicated. The introduction of $z$ leads to a conditional Gaussian prior distribution of $C$ which facilitates sampling from the full conditional distribution of $C$ for the grid boxes without proxy data and sequential updates of the grid boxes with proxy data. We do not see a simpler strategy which produces efficient proposal samples, when $z$ is marginalized (for example, a recent study shows that gradient based MCMC methods like Hamiltonian Monte Carlo are not faster than random walk Metropolis-Hastings algorithms for multi-modal problems (Mangoubi et al., 2018), which is intractable in our case due to the degrees of freedom of the posterior distribution).

Summarizing, we agree that $MC^3$ is a complex framework, for a model which looks

simple at first sight. However, we do not think that marginalization of $z$ is a simple solution because it only shifts the problem. Therefore, we do not see a simple modification of our MCMC algorithm to fit the kernel model, which would make the algorithm less complex. When the two other models suggested by the referee (see above) are fitted, the multi-modality of the posterior distribution vanishes such that a much simpler MCMC algorithm without parallel tempering can be used. This would indicate an additional advantage of those two models in the case of proper reconstruction performance of the two added models.

*"Maybe I missed it but what is the size of the model ensemble $K$ and the number of calibration sites?"*

The model ensemble has $K = 7$ members (implicitly stated in Sect. 2.2 and Table 1 of the manuscript). The regions that have been used for the transfer function calibration were determined separately for each taxa by pollen experts (Kühl et al., 2007). The number of calibration sites varies between 14.543 and 28.844, depending on the taxa. We will report these numbers in the revised manuscript.

*"How to you evaluate the Brier score when you don't include the absences? This seems to introduce a bias and could make the Brier score improper (which limitis its usefulness in comparing models)."*

We agree with the referee that using only occurring taxa to evaluate the Brier score is problematic and could make the Brier score improper when it is used to compare general models for predicting taxa presence and absence. However, our goal is an

indirect evaluation of predictions of past climate via transfer functions. In that context, it would lead to inconsistencies between the local reconstructions and the Brier score evaluations when we would include absences as there is currently no model available to accurately estimate detection probabilities for the reasons described above. In that context, inconsistencies mean that the local reconstructions could prefer systematically different climates than the Brier scores, when in one case absences would be included but not in the other case.

Therefore, we think that the evaluation would be much improved from future research to accurately estimate detection probabilities as this would allow the inclusion of present and absent taxa. But for the reasons described above, such an estimation is beyond the scope of this study. We think that the way how we use the Brier scores is still a useful technique to indirectly evaluate climate reconstructions. It should be noted that for each taxa the Brier scores are minimal for a unique climate state, but that minimum is bounded away from zero because the occurrence probability is bounded below one by the response surfaces. In addition, they are a convex function of the climate state for each taxa. We think that these two properties make our methodology useful for the comparison of climate reconstructions but it has to be noted that the comparison is conditioned on the correctness of the response surfaces.

Alternatively, predictions of past climate could be directly compared with probabilistic local reconstructions from the inversion of forward models, but this would mean that the local reconstruction have to be treated as (noisy) observations and not as an inferred product. Hence, we prefer to apply the forward model to the climate reconstructions and then compare the resulting predictions with observations in taxa space. This indirect strategy is also a recent way to infer skill of weather predictions.

In the revised manuscript, we will discuss the limitations of our evaluation methodology

more extensively and describe more detailed in which way it should be seen and interpreted. However, we do not plan to change the methodology because we do not see an easy way to fix its disadvantages for the reasons described above.

*"Scientific quality: Are the scientific approach and applied methods valid? Are the results discussed in an appropriate and balanced way (consideration of related work, including appropriate references)?*
*There are some questions about the implementation of the statistical model that are not completely resolved. (particularly the mixing distribution for climate doesn't make sense and the lack of absence data introduces bias in the estimates). The comments above can provide some guidance in resolving these issues."*

We hope that our changes according to the responses to the referee's comments above will improve the scientific quality of the revised manuscript.

*"Presentation quality: Are the scientific results and conclusions presented in a clear, concise, and well-structured way (number and quality of figures/tables, appropriate use of English language)?*
*The paper is reasonably well written from a technical perspective, although more motivation of why particular methods/equations are chosen would be useful. In other words, there is a lot written about what the methods are by not much about why the methods are chosen and what the ideas are trying to solve."*

We agree with the referee that more motivation for the statistical models and the respective inference algorithm would improve the quality of the manuscript substantially.

Therefore, we will explain our modelling choices in Sect. 3 (Methods) more extensively in the revised manuscript.

*"Are the scientific methods and assumptions valid and clearly outlined?*
*Not always (at least for the statistical methods)."*

We hope that our changes according to the responses to the referee's comments above will improve the validity of the scientific methods and assumptions as well as its presentation.

*"Does the title clearly reflect the contents of the paper?*
*Yes, with a small change of emphasis"*

The title of the revised manuscript will be changed according to the referee's suggestion.

*"Is the description of experiments and calculations sufficiently complete and precise to allow their reproduction by fellow scientists (traceability of results)?*
*For the most part. A gitHub repository/code base would go a long way."*
*"Is the amount and quality of supplementary material appropriate?*
*I would like to see more done for reproducibility. The computational methods seem overly complex and making code availalbe for replication would be useful."*

The revised manuscript will include paragraphs on data and code availability. We will create a repository to share our code. This repository will be referenced in the revised manuscript.

**References**

Anderson, J. L. and Anderson, S. L.: A Monte Carlo Implementation of the Nonlinear Filtering Problem to Produce Ensemble Assimilations and Forecasts, Monthly Weather Review, 127, 2741–2758, 1999.

Knutti, R., Furrer, R., Tebaldi, C., Cermak, J., and Meehl, G. A.: Challenges in combining projections from multiple climate models, Journal of Climate, 23, 2739–2758, 2010.

Krishnamurti, T. N., Kishtawal, C. M., LaRow, T. E., Bachiochi, D. R., Zhang, Z., Williford, C. E., Gadgil, S., and Surendran, S.: Improved Weather and Seasonal Climate Forecasts from Multimodel Superensemble, Science, 285, 1548–1550, 1999.

Kühl, N., Gebhardt, C., Litt, T., and Hense, A.: Probability Density Functions as Botanical-Climatological Transfer Functions for Climate Reconstruction, Quarternary Research, 58, 381–392, 2002.

Kühl, N., Litt, T., Schölzel, C., and Hense, A.: Eemian and Early Weichselian temperature and precipitation variability in northern Germany, Quarternary Science Reviews, 26, 3311–3317, 2007.

Liu, B., Ait-El-Fquih, B., and Hoteit, I.: Efficient Kernel-Based Ensemble Gaussian Mixture Filtering, Monthly Weather Review, 144, 781–800, 2016.

Mangoubi, O., Pillai, N. S., and Smith, A.: Does Hamiltonian Monte Carlo mix faster than a random walk on multimodal densities, arXiv:1808.03230v2, pp. 1–45, 2018.

Salter-Townshend, M. and Haslett, J.: Fast inversion of a flexible regression model for multivariate pollen counts data, Environmetrics, 23, 595–605, 2012.

Silverman, B.: Density Estimation for Statistics and Data Analysis, vol. 26 of *Monographs on Statistics and Applied Probability*, Chapman & Hall / CRC, Boca Raton, 1986.

Tawn, N. G. and Roberts, G. O.: Accelerating Parallel Tempering: Quantile Tempering Algorithm (QuanTA), arXiv:1808.10415v1, pp. 1–39, 2018.

[Figure]

[Figure]

Figure 1: Response surfaces for Betula nana (a) and Hedera helix (b). The ratio of presence versus absence in the modern calibration data in each bin with at least one data point is shown in colours. Gray bins are bins without data. The response surfaces (probability of presence according to Eq. (7) in the original manuscript) are depicted by contour lines. In the climate space, combinations of MTWA and MTCO above a line at MTWA = MTCO cannot occur by definition. The white triangle in the upper left are artificial absence information added to account for this constraint, as described in the original manuscript.

**Fig. 1.**

---

## Author Comment (AC2) · 10 Dec 2018

We thank the referee for taking the time to review our discussion paper, and for helpful and interesting comments which will improve the quality of our manuscript. As a response to the referee's suggestions, we plan substantial changes of the manuscript. In the following, referee's comments (RC) are given in blue italicised text and followed by our respective responses (AR).

*"I have some minor criticism of the presentation which often appears poorly-organised*

*to me. Specifically, many parameters and concepts are used prior to their definition and introduction which made the manuscript tough going for me."*

We will go through the manuscript and try to rework Sect. 2 and 3 to improve the structure of the manuscript.

*"A more fundamental concern I have is demonstrated in the results, which (unless I have misunderstood something) show a strong degree of instability and overconfidence. This is a common occurrence in situations where all members of a model ensemble are inadequate and a Bayesian framework is used which does not allow for this - for instance, by the explicit use of model inadequacy or model error terms. The z weight of 0.98 when all data are used is itself likely to indicate a problem, and this feeling is only strengthened by the fact that the weight on this model was essentially 0 when half the data were used, in which event two different models are preferred. What seems to be happening here is that the method is homing in on the model(s) for which the data are most likely, while in fact these data are incompatible with any model.*
*I've seen this in a variety of contexts and am sure this sort of phenomenon will be familiar to the authors. A trivial demonstration of the phenomenon can be given by considering a situation in which we have two models which generate data according to N(0,1) and N(10,1) respectively, whereas the data are in fact generated by N(5,1). A naive filtering which ignores the possibility of model inadequacy will assign essentially all weight to whichever model happens to be closer to the sample mean of the data set (which is of course close to, but not precisely, 5). As more data are sampled, the weight will jump randomly from one model to the other with extremely high confidence, rather than converging to the answer that the models are equiprobable but poor, which might be found if a reasonable model error term was considered. This issue seems to me to undermine the results and details of the application in a*

*fairly fundamental manner but I would hope that the authors could revise their method to adequately account for it, ie by accounting for model inadequacy in a formal manner."*

We agree with the referee that the ensemble member weights in our model tend to degenerate towards the least wrong model and this degeneracy increases with increasing signal in the proxy data. This is a general problem of particle filter type models (or more general, Bayesian model selection problems). For that reason, we combine the particle part of our model with a part that is more similar to Kalman type filters by adjusting the ensemble members according to a prior covariance and the signal in the proxy data.

On the other hand, the strong changes of model weights between the joint reconstruction and the MTCO only reconstruction are a result of a different proxy data calibration and not of leaving out half the data. The proxy data in the MTCO only reconstruction is still the same but instead of calibrating it against a bivariate climate, it is only calibrated against MTCO. It is not surprising that different models perform better for winter than for summer climate. The reason that the joint reconstruction is dominated by the model performance for summer climate is a result of the higher signal to noise ratio for local MTWA reconstructions because most taxa are more sensitive to temperature during the growing season than in winter. We briefly report in Sect. 5.1 results from experiments where half of the proxy data is left out. In each of these results the same model than in the reconstruction with the full dataset is preferred, but because of the smaller signal in the proxy data the weights can be less degenerate (see Fig. at the end of this response). Moreover, the experiments show that the reconstructions are reasonably stable with respect to leaving out substantial parts of the data. We think that the issue of very different model weights between the joint reconstruction and the MTCO only reconstruction rather indicates that the joint reconstruction could be improved from different weights for MTWA and MTCO, even though this would reduce

the physical consistency of the estimates.

Nevertheless, we agree with the referee that our model produces overconfident esti-
mates and an under-dispersed posterior distribution similar to many ensemble filtering
models, and that this could be reduced by techniques to formally account for model
inadequacy. The main reason for this behaviour is the small number of ensemble
members leading to a small number of mixture kernels and a small number of spatial
modes in the covariance matrix $\Sigma$. Accounting for model inadequacy in a physically
reasonable way is a challenging issue and an active research area particularly in
postprocessing applications. To reduce the underdispersion of our model and reduce
reconstruction biases, we introduce two extensions of the model compared to the
original manuscript.

First, we compare the proposed kernel model with a statistical model that facilitates a
more flexible prior mean structure by mixing directly over $\omega_k \mu_k$, which was suggested
by the other referee (see also response to Referee 1). This means that weighted
averages of the ensemble member climatologies are used and the weights are
adjusted according to the proxy data. Thereby, we increase the degrees of freedom
but ensure that the spatial structures are still physically motivated. In particular, the
model weights are less degenerate with this adjustment and the model can better
cope with situations like the one described by the referee where the truth is located
between two kernels.

Second, we compare the covariance matrix regularized by the glasso algorithm
with a shrinkage matrix (Hannart and Naveau, 2014), where the empirical corre-
lation matrix of the ensemble is combined with a Matérn type correlation matrix.
This increases the spatial modes of the prior covariance matrix and estimating
the shrinkage weight from the proxy data ensures that the amount of deviations

from the ensemble correlation matrix is in agreement with the proxy data. Initial results of identical twin experiments reveal that this increase of modes in the prior covariance matrix reduces the underdispersion of the posterior distribution significantly.

Combining ensemble filtering methods with additional techniques to account for model inadequacy should be an important part of future work on climate field reconstructions as a balance has to be found between underdispersion of the posterior distribution and overfitting to noisy proxy data by enhancing the degrees of freedom too much. Beyond our adjustments compared to the original manuscript, some directions that can be envisaged are the increase of permitted spatial structures in the prior mean by combining the climate simulation ensemble with patterns calculated from alternative physically motivated models, and the introduction of multiple shrinkage targets (Gray et al., 2018) in the prior covariance matrix which allows the proxy data to weight multiple spatial correlation modes. Identical twin experiments and cross-validation experiments with proxy compilations will be important techniques to understand the appropriateness of different modelling options as described below.

In the revised manuscript, we will extend the report on results from experiments with reduced data as a way to analyse the robustness of our reconstructions. In addition, we will include the two adjustments in the model described above in the methods section. Afterwards, we will add a section that compares these adjusted models with the original kernel method and the other alternative models described in the response to Referee 1. This section will include identical twin and cross-validation experiments as described below. Finally, we will put a focus in the discussion of future research strategies on additional methods to account for model inadequacy.

*"Related to this, the Brier score analysis seems to indicate that the posterior is*

*generally closer to the data than the prior, but it does not indicate whether the posterior is valid in the sense of having reasonably calibrated uncertainties. Also, while the data are in taxa-space, the aim of the paper is actually to recreate a climate, so it is important that the validity of this estimate is assessed. Perhaps a cross-validation (in which one model is used as a target) could be tried."*

We agree with the referee that ideally we want to evaluate the ability of our method to reconstruct climate. However, there are no direct observations of paleoclimate available such that evaluations against real observations have to be indirect. Therefore, evaluations with cross-validation experiments against indirect observations are a valuable tool to analyse the ability of reconstruction methods to reconstruct past climate. In particular, evaluating the reconstruction in taxa space by applying the forward model is a methodologically more stringent method than comparing reconstructions with other inferred quantities like local climate reconstructions. Therefore, for the evaluation of a reconstruction against observations, we do not see a way around evaluations against indirect data. This is also a recent way of model skill evaluation in weather forecasting.

The referee suggests to perform identical twin experiments, i.e. experiments in which one model is excluded from the ensemble as reference climatology and the goal is to reconstruct this reference climatology from the remaining models. This is a useful method to study the validity of the posterior distributions which is difficult to analyse in evaluations against observations. Therefore, we designed such experiments by simulating pseudo-proxies from the reference climatology using the proxy uncertainty structure from the local reconstructions, reconstructing the reference climatology from the pseudo-proxies with the model proposed in the original manuscript as well as alternative statistical models proposed by the referees, and analysing the skill of the corresponding posterior distributions to predict the reference climatology. In particular, this methodology is well-suited for studying potential underdispersion of the posterior

distributions and for comparing different statistical models.

Initial results from these experiments suggest that the models with the shrinkage matrix are substantially less under-dispersed than the models with the glasso optimized covariance matrices as a result of containing more spatial modes and therefore more degrees of freedom. Moreover, the kernel model that we proposed in the original manuscript performs similar to the purely Gaussian model and the mixture over $\omega_k \mu_k$ which were suggested by the first referee. Therefore, it seems like adding spatial modes to the covariance structure results in the biggest improvements of reconstruction skill.

In the revised manuscript, we will add a description of the design of identical twin experiments, and report on the results of these experiments. In addition, we will dedicate a section to the comparison of the different statistical models, the one that is proposed in the original manuscript as well as alternatives that were suggested by the referees, using identical twin experiments as well as cross-validations in taxa space.

*"However, I am not convinced that the details of the application as presented here are adequate, nor, therefore, can I believe that their results are credible. I urge them to modify and extend their work in order to address this."*

As described above, we will extend our work by addressing the comments and include the new results in the revised manuscript. We hope that this will increase the credibility of our results.

*"As I mentioned, the ordering of some of the paper made it hard work for me.
The prior is introduced in Sect 2.2 and Fig 1 but only explained in Sect 3.2. The variables w and z are first mentioned in 3.1, but for their definitions we are referred forward to 3.2. It is only after substantial usage of the variables, right at the end of 3.2, that we are informed that z was only introduced as a help parameter, with its definition and explanation again pushed forward another two sections to 3.4 with the largely unrelated discussion of the transfer function (two words for this please, and note also that burn-in should be hyphenated) intervening in section 3.3. I would have found it far more straightforward to have had the variables explained as they were introduced.
The multinomial with n=1 would I think be better described as the categorical distribution.
In 3.4, "alternately" usually refers to two alternates, not a sequence. "Sequentially" might be better."*

We thank the referee for pointing out several issues that hamper the readability of our manuscript. We will reorder Sect. 2 and 3 according to the referee's suggestions and incorporate the linguistic advises.

**References**

Gray, H., Leday, G. G., Vallejos, C. A., and Richardson, S.: Shrinkage estimation of large covariance matrices using multiple shrinkage targets, arXiv:1809.08024v1, pp. 1–32, 2018.
Hannart, A. and Naveau, P.: Estimating high dimensional covariance matrices: A new look at the Gaussian conjugate framework, Journal of Multivariate Analysis, 131, 149–162, 2014.

[Figure]

Figure 1: Posterior ensemble member weights (mean of $z$) of the the joint reconstruction and of five experiments where half of the proxy samples are left out. The black diamonds correspond to the reconstruction with the full dataset, whereas the coloured diamonds represent the experiments with reduced data. Prior weights (mean of $z$) are denoted by the dashed line.

**Fig. 1.**

---

## Referee Report (RR1)

**Review: Combining a pollen and macrofossil synthesis and climate simulations for spatial reconstructions of European climate using Bayesian filtering**

I appreciate the effort of the authors to address the comments of their manuscript in the revision. I think this revision is thoughtful and useful. I still have reservations about the interpretation of the statistics with specific comments below. I think the issues can be easily addressed with either small changes in the modeling or modifications in the writing.

I **very strongly** thank the authors for the inclusion of their code on Bitbucket and applaud them on their time putting this together. This is a major, and important, undertaking and is greatly appreciated.

- I find the fact that the regularization of the covariance matrix influences prediction more than model choice to be concerning (page 19, lines 30-32). I understand that this is common in the literature; however, I do think the difference in prediction based on regularization method suggest that the covariance matrix is not necessarily representing what it is supposed to. This is especially true given the effects of regularization suggest that $\Sigma$ isn't really representing ensemble variability - just prior assumptions about the covariance structure. More of a discussion about how the impact of the regularization implies the interpretation of what $\Sigma$ isn't as clear as inter-model variability even though in the ideal world this is what $\Sigma$ would represent.

- page 20, lines 29-30 - significant if the **posterior** probability to exceed the...

- Page 8, line 21 - The prior for $\theta$ of $N(0, 10)$ is somewhat too vague for a logistic regression (in fact, it is actually somewhat informative). For example, it puts most of the prior mass on 0 or 1 and little in between. A $N(0, 2)$ is often a better choice for logistic models.

```
library(LaplacesDemon)
layout(matrix(1:2, 1, 2))
**Normal (0, 10)**
theta <- rnorm(1000, 0, 10)
p <- invlogit(theta)
hist(p, main = "N(0, 10) prior")
**Normal (0, 2)**
theta <- rnorm(1000, 0, 2)
p <- invlogit(theta)
hist(p, main = "N(0, 2) prior")
```

| **N(0, 10) prior** | **N(0, 2) prior** |
|---|---|

[Figure]

[Figure]

- Equation 8 - Why the $(1/2 \, , \ldots \, 1/2)$? A $(1, \ldots, 1)$ prior produces an *a priori* uniform distribution over mixtures. I believe a $(1/2, \ldots, 1/2)$ prior pushes the weights more towards the extremes of the composition which could influence the collapsing of the weights to a small number of ensemble members. A better choice would be to use an $\alpha$ $(1, \ldots, 1)$ prior and assign $\alpha$ some prior (e.g., a gamma(1, 1)). Notice how in the first plot, there are more samples at the verices of the plot suggesting a collapsing of weights. A more relaxed prior might also improve MCMC mixing.

```r
library(ggtern)
```

```
**Loading required package: ggplot2**
```

```
**--**
**Remember to cite, run citation(package = 'ggtern') for further info.**
**--**
```

```
##
**Attaching package: 'ggtern'**
```

```
**The following objects are masked from 'package:ggplot2':**
##
**%+%, aes, annotate, calc_element, ggplot, ggplot_build,**
**ggplot_gtable, ggplotGrob, ggsave, layer_data, theme,**
**theme_bw, theme_classic, theme_dark, theme_gray, theme_light,**
**theme_linedraw, theme_minimal, theme_void**
```

```r
N <- 500
draws <- rbind(LaplacesDemon::rdirichlet(N, 0.5 * c(1,1,1)),
               LaplacesDemon::rdirichlet(N, c(1,1,1)),
               LaplacesDemon::rdirichlet(N, 2 * c(1,1,1)))
ggtern(data = data.frame(x = draws[, 1],
                         y = draws[, 2],
                         z = draws[, 3],
```

```
                            alpha = rep(c(0.5, 1, 2), each = N)),
      aes(x, y, z)) +
facet_wrap( ~ alpha) +
geom_point(alpha = 0.25) +
theme(legend.position = "NULL") +
theme_bw()
```

[Figure]

- Equation 21 - Skill scores in general are not proper scoring rules. I know these are common in the literature, but I find the widespread use to be concerning, especially where there are many alternatives that convey the same information but don't lose propriety (e.g., just present the scores, present the difference in scores (not scaled by BS(prior)), etc.).

---

## Author Response (AR2)

**Reply to comments by the referees and the editor**

We thank the referees and the editor for reviewing the revised version of our manuscript, and for helpful comments to improve our manuscript. We revised our manuscript with respect to the suggestions of the first referee and the editor. In the following, referee's comments (RC) and editor's comments (EC) are given in blue italicized text and followed by our respective responses (AR). Finally, we list all changes to the manuscript.

**Response to the first referee's comments**

*" I find the fact that the regularization of the covariance matrix influences prediction more than model choice to be concerning (page 19, lines 30-32). I understand that this is common in the literature; however, I do think the difference in prediction based on regularization method suggest that the covariance matrix is not necessarily representing what it is supposed to. This is especially true given the effects of regularization suggest that $\Sigma$ isn't really representing ensemble variability - just prior assumptions about the covariance structure. More of a discussion about how the impact of the regularization implies the interpretation of what $\Sigma$ isn't as clear as inter-model variability even though in the ideal world this is what $\Sigma$ would represent."*

We suspect that the main reason for the strong effect of the regularization on the reconstruction performance is the small ensemble size. Hopefully, a larger number of simulations with sufficient resolution will become available in the next years, for example within PMIP4, such that this hypothesis can be tested. Additionally, multiple ensemble members from the same ESM will allow an examination of internal model variability versus inter-model variability.
Additionally, the two versions for regularization are two relatively extreme cases with the glasso leading to a small number of effective degrees of freedom versus a much larger number for the shrinkage approach. That the glasso approach exhibits under-dispersive behavior while the shrinkage approach exhibits over-dispersive behavior in the identical twin experiments suggests that, at least for these experiments, the optimal number of degrees of freedom lies between those two cases. This is particularly true considering that it is unreasonable to assume that the modes from the inter-model variability are the optimal modes to explain the spatial variability of the true climate state (as it is for all models of high-dimensional complex systems). However, it is unclear how this translates to real-world situations such that an optimization procedure for the number of spatial modes in $\Sigma$ is not straightforward and beyond the scope of our study. As discussed in Sect. 5.3, improving the model for $\Sigma$ will be an important task for future research. In the revised manuscript, we extend the discussion on the interpretation and potential shortcomings of $\Sigma$ in Sect. 4.1.3 and 5.3. We hope that this clarifies the reason for the strong effect of the regularization technique on the reconstructions, and helps designing future studies.

*"page 20, lines 29-30 - significant if the **posterior** probability to exceed the. . ."*

We added the word 'posterior' in the manuscript.

*"Page 8, line 21 - The prior for θ of N(0, 10) is somewhat too vague for a logistic regression (in fact, it is actually somewhat informative). For example, it puts most of the prior mass on 0 or 1 and little in between. A N(0, 2) is often a better choice for logistic models."*

We checked the effect of using a $\mathcal{N}(0, 2)$ prior instead of $\mathcal{N}(0, 10)$ and compared the results with a maximum likelihood fit for the coefficients. For most taxa, the difference between the three models is negligible due to the high information content of the calibration data. For rare taxa, slightly less decisive response surfaces (i.e. less mass near 0 or 1) are found for the $\mathcal{N}(0, 10)$ prior than for the maximum likelihood estimate. Similarly, the response surfaces with $\mathcal{N}(0, 2)$ prior are slightly less decisive than those with $\mathcal{N}(0, 10)$ prior. The effect of the different prior on the local reconstructions and the spatial reconstructions is insignificant and we could not diagnose an improvement in prediction quality for the narrower prior. Therefore, we added a motivation for the choice of prior distribution in the manuscript, but did not change our model and the results presented in the manuscript.

*"Equation 8 - Why the (1/2 , . . . 1/2)? A (1, . . . , 1) prior produces an a priori uniform distribution over mixtures. I believe a (1/2, . . . , 1/2) prior pushes the weights more towards the extremes of the composition which could influence the collapsing of the weights to a small number of ensemble members. A better choice would be to use an γ (1, . . . , 1) prior and assign α some prior (e.g., a gamma(1, 1)). Notice how in the first plot, there are more samples at the verices of the plot suggesting a collapsing of weights. A more relaxed prior might also improve MCMC mixing."*

The Dir(1/2,...,1/2) gives equal weight to each ensemble member, but in each MCMC step one model tends to dominate the regression because comparably strong weights are given to extreme states. The motivation behind this prior is that it preserves relatively much structure from one ensemble member, which produces a physically more consistent estimate. However, the weighted averaging of the models allows to correct for climate model inadequacies of the dominant ensemble member using the other ensemble members. In that sense, the Dir(1/2,...,1/2) is also a compromise between the kernel model, which preserves more physical consistency by choosing a single ensemble member in each MCMC step, and the Gaussian model, which gives equal weight to all ensemble members, and thereby removing physical consistency.
Using Dir(1,...,1) and Dir(2,...,2) prior distributions instead, we found more balanced ensemble member weights (the higher the parameter is chosen, the more balanced). This leads to reconstruction results closer to the Gaussian model, which exhibits similar reconstruction skill to the regression model and does not exhibit significant reconstruction differences compared to the regression model (compare with manuscript Supplement).
Using $\alpha$ to estimate the weight dispersion from the proxy data is an elegant idea for future research. Implementing this extension of our model will require substantial testing effort to study the effect of the variable parameter. Since we do not expect improved reconstruction skill from it as described above, we do not implement it in our model for the revised manuscript, and leave this development to future research.
In the revised manuscript, we expand our motivation for using the Dir(1/2,...,1/2) prior in Sect. 3.3.2. In addition, we acknowledge the idea of a variable hyperprior $\alpha$ by mentioning it in Sect. 5.3 of the revised manuscript.

*"Equation 21 - Skill scores in general are not proper scoring rules. I know these are common in the literature, but I find the widespread use to be concerning, especially where there are many alternatives that convey the same information but don't lose propriety (e.g., just present the scores, present the difference in scores (not scaled by BS(prior)), etc.)."*

We acknowledge that skill scores are not proper and therefore not optimal to describe the reconstruction skill. However, we think that they are useful to compute the added value of the reconstructions compared to the unconstrained PMIP3 ensemble. In particular, they are commonly used in the respective literature and therefore allow the general audience to easily understand the added value of the reconstruction. Therefore, we retain the skill scores in the current form in the revised manuscript. Advancing the methodology for comparison of climate simulations and proxy data with respect to the involved uncertainties is an ongoing research topic and we hope that these developments will also lead to improved scoring rules for the comparison of reconstruction methods.

**Response to the editor's comments**

*"I just ask you to check whether you can shorten the manuscript a little (say, by about 10 %) without much effort, and to closely follow up on the final comments by the first referee."*

The changes in response to the comments of the first referee are described above. We shortened the manuscript by approximately 10% without removing any substantial parts of the contents.

*"Please consider carefully whether you only want to modify the manuscript regarding the thoughtful choice of prior distributions, or you think it pays off in terms of improvements in the model results when you change your code as well."*

As described above, we enhanced our discussion for the choice of priors. As no change of reconstruction skill and no significant change of reconstruction results was diagnosed, we did not change the results part of the manuscript and the Bitbucket repository.

**Changes to the manuscript**

- We updated the order of the affiliations and the email address of the correspondence author.

- In Sect. 3.2 (Transfer function), we added a motivation for the choice of parameter prior distributions, and expanded on the effect of the prior on the estimates.

- In Sect. 3.3.2 (Regression model), we motivate the choice for the prior distribution of $\lambda$.

- In Sect. 3.3.4 (Glasso based covariance matrices), we renamed the penalty parameter (from $\rho$ to $\xi$) to avoid confusion with the range parameter $\rho$ in the shrinkage approach.

- In Sect. 4.1.3 (Conclusions from the comparison study), we enhanced our interpretation of the strong effect of the covariance matrix regularization technique on the reconstruction skill (compared to the process stage formulation).

- In Sect. 5.3 (Climate model inadequacy and process stage structure), we expanded the discussion of the spatial covariance matrix choice and the effect of regularization techniques. In addition, we added a discussion of a possible extension of the Bayesian model by a hyperprior to control the dispersion of ensemble member weights.

- We shortened Sect. 5.3 (Climate model inadequacy and process stage structure) by removing paragraphs that discuss more general extensions of our model to focus on direct solutions to potential shortcomings of the framework.

- p. 18, l. 10: We added the word 'posterior' for clarification.

- We shortened many paragraphs to reduce the length of the manuscript by around 10% without removing significant parts of the contents.

- Several corrected typos, and small changes to improve clarity and readability of the manuscript are not listed here explicitly, but can be seen in the marked up version of the manuscript.

[revised manuscript text omitted]
  response functions is negligible for most taxa. It slightly smooths the corresponding maximum likelihood estimates particularly for rare taxa, but does not influence the reconstructions significantly.

[revised manuscript text omitted]

$$\Sigma_{\text{emp}} \;=\; \frac{1}{K-1} \sum_{k=1}^{K} \left( \mu_k - \bar{\mu} \right) \left( \mu_k - \bar{\mu} \right)^t. \tag{6}$$

The superscript $t$ denotes the matrix transpose. Hence, the covariance matrix is based on the inter-model differences as an estimate of epistemic uncertainties. The regularization techniques of $\Sigma_{\text{emp}}$ are specified below. The Gaussian distribution is  $N$ dimensional, where $N$ is the number of grid boxes times the number of jointly reconstructed variables.

The main advantage of this Gaussian model (GM) is that inference  is simpler than in more complex probability density estimation techniques. The disadvantage is that it relies on the strong assumption that  $\mu_k$ are iid samples from an unknown Gaussian distribution. This assumption tends to be more realistic for samples from just one ESM, whereas statistics of multi-model ensembles are often not well described by purely Gaussian distributions (Knutti et al., 2010). A second disadvantage of this model is that the absence of additional parameters limits the possibilities to adjust the posterior distribution to the proxy data. The third disadvantage  is that spatial structures of  individual ensemble members are lost by averaging over all  members. Nevertheless, in many climate prediction applications multi-model averages outperformed each individual ESM (e.g. Krishnamurti et al., 1999).

**3.3.2 Regression model**

A relaxation of the assumptions of the GM is the second model, that we call the regression model (RM) because it is inspired by regression based models popular in postprocessing and climate change detection and attribution (Hegerl and Zwiers, 2011). In the RM,  variable weights $\lambda_k, k = 1, ..., K$ are introduced to allow weighted averages of the ensemble members. This means, that samples, which fit better to the proxy data, are weighted higher in the posterior. The sum of the weights is set to one such that unrealistically warm or cold state are prevented. This leads to the process stage model

$$\mathbb{P}(C_p | \lambda_1, ..., \lambda_K) = \mathcal{N}\left( C_p \,\middle|\, \sum_{k=1}^{K} \lambda_k \mu_k, \Sigma_{\text{prior}} \right), \tag{7}$$

and an additional prior distribution for the model weights

$$\mathbb{P}(\lambda) = \text{Dir}\left( \lambda_1, ..., \lambda_K \,\middle|\, \tfrac{1}{2}, ..., \tfrac{1}{2} \right). \tag{8}$$

Dir denotes a Dirichlet distribution, which  guarantees that the weights take values between zero and one and sum up to one. Conditioned on  $\lambda$, the process stage distribution is Gaussian, but non-Gaussianity is permitted through the variable weights.

 The parameters of the prior distribution are chosen to prefer combinations with one dominant ensemble member which is adjusted by the other members to improve the fit to the proxy data. Thereby, it preserves more physical structure from individual members than the GM as better fitting models are only weakly corrected for climate model inadequacies. The RM has the advantage of possessing more degrees of freedom compared to the GM. The inference process becomes a little more involving than for the GM because the ensemble member weights have to be estimated, too, but the conditional Gaussian distribution of $C_p$ helps designing efficient inference algorithms.

**3.3.3 Kernel model**

The third model has been introduced in the data assimilation literature by Anderson and Anderson (1999) to combine particle and Gaussian filtering approaches. This kernel model (KM) assumes that each ensemble member is a sample from an unknown distribution of possible climate states given a set of forcings, but it does not assume that this unknown distribution is Gaussian. Instead, non-parametric kernel density estimation techniques (Silverman, 1986), where the probability distribution is given by a mixture of multivariate Gaussian kernels, are used. Each ensemble member climatology corresponds to the mean of a kernel.

Ideally, the covariance matrix of each kernel would correspond to the respective ESM, such that the spatial autocorrelation of that ESM is preserved when we sample from its kernel. Unfortunately, there is only one MH run available for each ESM and the internal variability in those runs is much smaller than the inter-model differences. Using the internal variability of

those runs would thus lead to very distinct kernels and allow too few climate states. Therefore, the covariance of each kernel is estimated from the inter-model differences  even though autocorrelation of the individual models is lost. This  is a very common choice in kernel based probability density approximations  (Liu et al., 2016; Silverman, 1986).

Compared to the GM, the empirical covariance matrix $\Sigma_{\text{emp}}$ is scaled by the Silverman factor (Silverman, 1986)

$$f := \left( \frac{4}{K \cdot (N+2)} \right)^{\frac{2}{N+4}}, \tag{9}$$

which optimizes the variances of the kernels. Hence, in the KM the scaled empirical covariance matrix $\tilde{\Sigma}_{\text{emp}}$, given by $f \cdot \tilde{\Sigma}_{\text{prior}}$, is regularized leading to the spatial covariance matrix $\tilde{\Sigma}_{\text{prior}}$. Note that the small number of ensemble members   leads to a standard deviation reduction of only around 2% in our applications.

Each kernel gets an assigned weight $\omega_k$, $k = 1, ..., K$, which is inferred in the Bayesian framework. The weights sum up to one. The resulting process stage is a mixture distribution

$$\mathbb{P}(C_p \,|\, \omega_1, ..., \omega_K) = \sum_{k=1}^{K} \omega_k \, \mathcal{N}\left( C_p \,|\, \mu_k, \tilde{\Sigma}_{\text{prior}} \right). \tag{10}$$

A Dirichlet distributed prior is used for $\omega$ with parameter $\frac{1}{2}$ for each of the $K$ components. A computational disadvantage of the KM is that the process stage is multi-modal and non-Gaussian. We augment the model by an additional parameter $z$, which follows a categorical distribution, denoted by Cat, to restore that $C_p$ is Gaussian conditioned on the $\omega$ and $z$. $z$ selects a kernel $k$ according to its weight $\omega_k$, i.e. $z$ is defined such that

$$\mathbb{P}(\omega) = \text{Dir}\left( \omega_1, ..., \omega_K \,\Big|\, \tfrac{1}{2}, ..., \tfrac{1}{2} \right) \tag{11}$$

$$\mathbb{P}(z \,|\, \omega) = \text{Cat}\left( z_1, ..., z_K \,|\, \omega_1, ..., \omega_K \right) \tag{12}$$

$$\mathbb{P}(C_p \,|\, z) = \prod_{k=1}^{K} \left( \mathcal{N}(C_p \,|\, \mu_k, \Sigma_{\text{prior}}) \right)^{z_k}. \tag{13}$$

Integrating out $z$ yields the mixture distribution Eq. (10).

Two advantages of the KM are that it is not assumed that the unknown prior distribution is Gaussian and that the kernels do not rely on an iid assumption for their first moment properties. However, the KM still relies on an iid assumption for the second  moments. The KM preserves the spatial structures of each ESM in the first moments of the kernels . This preservation of physical consistency reduces the degrees of freedom compared to the RM. For example, when the true climate state lies exactly between $\mu_1$ and $\mu_2$  the posterior mode cannot be changed to $\frac{1}{2}(\mu_1 + \mu_2)$, which is possible in the RM. Another disadvantage of the KM is that the multi-modality makes the design of efficient inference algorithms a lot more challenging.

**3.3.4 Glasso based covariance matrices**

The first technique to regularize the empirical covariance matrix (the scaled empirical covariance in the KM), which is applied in this study, is the graphical lasso algorithm (glasso, Friedman et al., 2008, implemented in the R-package glasso). This algorithm approximates the precision matrix (inverse covariance) by a positive definite, symmetric, and sparse matrix $\Sigma_{\mathrm{prior}}^{-1}$. Therefore, $\Sigma_{\mathrm{prior}}$ is a valid $N$-dimensional covariance matrix. Glasso maximizes the penalized log-likelihood

$$\log \det \Sigma_{\mathrm{prior}}^{-1} - \mathrm{trace}(\Sigma_{\mathrm{emp}}\,\Sigma_{\mathrm{prior}}^{-1}) - \underline{\rho}\,\xi\,\|\Sigma_{\mathrm{prior}}^{-1}\|_1, \tag{14}$$

[revised manuscript text omitted]

 The strong effect of the covariance regularization technique on the reconstructions might originate from the small ensemble size. This hypothesis can be tested when more simulations with sufficient resolution become available for example from the PMIP4 project. In addition, it indicates that the modes of the empirical covariance matrix do not optimally explain the spatial variability of the climate and the corresponding uncertainty structures. The difference between under-dispersive behavior in ITEs with glasso models and over-dispersion for shrinkage models suggests that the optimal number of effective degrees of freedom lies between those two models. However, an optimization procedure for the number of spatial modes in the covariance matrix is not straightforward and left for future research.

In the current study, we use a fixed prior distribution for the ensemble member weights

~~The implementation of these different types of process stages would facilitate more quantitative comparisons of reconstructions and allow a fair testing of modelling assumptions by using ITEs and CVEs. In particular, the reformulation of existing climate field reconstruction techniques as BHM (Tingley et al., 2012) offers many model comparison techniques that are currently not available as the borders between very different statistical techniques especially to estimate reconstruction uncertainties have to be crossed~~(compare with Sect. 3.3). An extension of this model would be to let the proxy data inform whether more balanced weights should be favored or weights with one dominant ensemble member. This can be achieved by using a hyperprior that controls the concentration of the weights.

**6 Conclusions**

We presented a new method for probabilistic spatial reconstructions of paleoclimate. The approach combines the strengths of pollen  records, which provide information about the local climate state, and of climate simulations, which downscale forcing conditions to physically consistent regional climate patterns. Thus, we reconstruct physically reasonable spatial fields, which are consistent with a given proxy synthesis. Our framework can deal with probabilistic transfer functions, which are non-linear and non-Gaussian, such that an extension to a wide range of proxies and associated transfer functions is possible.

Using ITEs and CVEs, we showed that robust spatial reconstructions with Bayesian filtering methods that exhibit small biases and are not under-dispersed are possible as long as the statistical framework is flexible enough to account for deficiencies of climate simulations and to avoid filter degeneracy, which can emerge due to small ensemble sizes and biases in climate simulations.  The resulting model, which is used for spatial reconstructions of European MH climate, uses a weighted average of the involved ensemble member climatologies and a shrinkage matrix approach for spatial interpolation and structural extrapolation of the proxy data.

We apply our framework to reconstruct MTWA and MTCO in Europe during the MH using the proxy synthesis of Simonis et al. (2012) and the PMIP3 MH ensemble. Brier scores from cross-validations reveal that the spatial reconstruction predominantly adds value to the unconstrained PMIP3 ensemble, and analyses of the spatial homogeneity of the posterior distribution indicate a reasonable degree of spatial smoothing. The large scale spatial patterns of our reconstruction are in agreement with previous work (Mauri et al., 2015; Bartlein et al., 2011). As the posterior mean is more similar to the local proxy reconstructions than to the prior mean for most terrestrial areas, we see that  a reconstruction, which is in line with reconstructions that do not include simulation output, is possible despite well-known model-data mismatches (Mauri et al.,

2014). Our framework provides a way to quantitatively test hypotheses in paleoclimatology and to assess the consistency of a given proxy synthesis.

*Code and data availability.* R code for computing reconstructions with the presented Bayesian framework is provided in a Bitbucket repos-
5   itory available under https://bitbucket.org/nils_weitzel/spatial_reconstr_repo. The pollen and macrofossil synthesis is published in Simonis (2009). It is available in the Bitbucket repository. The PMIP3 MH simulations are available in the CMIP5 archives. In this study, they were downloaded from the DKRZ long term archive CERA (https://cera-www.dkrz.de). The modern climate data was downloaded from the University of East Anglia Climatic Research Unit, available at http://www.cru.uea.ac.uk/data/. The vegetation data for transfer function calibration was provided by Thomas Litt and Norbert Kühl.

10  **Appendix A: Determination of glasso penalty parameter**

To determine the glasso penalty parameter $\xi$, we first recognize that for values smaller than  $\xi = 0.3$ the resulting matrices become numerically unstable in our application due to the small ensemble size. Five values for  $\xi$ were tested: 0.3, 0.5, 0.7, 1.0, and 2.0. Larger values lead to sparser precision matrices and therefore to  weaker spatial correlations.  For each of the five parameters, we
15  perform CVEs with the RM and compare the resulting BS (see Sect. 4.1.2). While the smallest penalty parameters have the best mean BS, the differences are generally small (see Supplement). However, the influence of the penalty term in Eq. (14) on the overall regression increases from 79.5% for  $\xi = 0.3$ to 98.5% for $\xi = 2.0$. Based on these diagnostics, we choose  $\xi = 0.3$ for the reconstructions in this study. 
[revised manuscript text omitted]

[Figure]

**Figure 3.** Response functions for  (a) Betula nana and  (b) Hedera helix. The relative frequency of occurrence in 1K bins is shown in colors, and the contours depict the probability of presence as estimated by the logistic response function. Gray boxes denote bins without calibration data. In the climate space, combinations of MTWA and MTCO with MTWA < MTCO cannot occur by definition. White bins in the upper left depict artificial absence information added to account for this constraint.

[revised manuscript text omitted]